# ScienceAgentBench:
# Toward Rigorous Assessment of Language Agents for Data-Driven Scientific Discovery

**Ziru Chen**[1*]**, Shijie Chen**[1*]**, Yuting Ning**[1]**, Qianheng Zhang**[3]**, Boshi Wang**[1]**, Botao Yu**[1]**,
**Yifei Li**[1]**, Zeyi Liao**[1]**, Chen Wei**[3]**, Zitong Lu**[4]**, Vishal Dey**[1]**, Mingyi Xue**[5]**,
**Frazier N. Baker**[1,6]**, Benjamin Burns**[1]**, Daniel Adu-Ampratwum**[2]**, Xuhui Huang**[5]**,
**Xia Ning**[1,2,6]**, Song Gao**[3]**, Yu Su**[1]**, Huan Sun**[1*]

[1]Department of Computer Science and Engineering, OSU    [2]College of Pharmacy, OSU
[3]Department of Geography, UW–Madison    [4]Department of Psychology, OSU
[5]Department of Chemistry, UW–Madison    [6]Department of Biomedical Informatics, OSU
Website: https://osu-nlp-group.github.io/ScienceAgentBench/

## Abstract

The advancements of large language models (LLMs) have piqued growing interest in developing LLM-based language agents to automate scientific discovery end-to-end, which has sparked both excitement and skepticism about the true capabilities of such agents. In this work, we argue that for an agent to fully automate scientific discovery, it must be able to complete all essential tasks in the workflow. Thus, we call for rigorous assessment of agents on individual tasks in a scientific workflow before making bold claims on end-to-end automation. To this end, we present ScienceAgentBench, a new benchmark for evaluating language agents for data-driven scientific discovery. To ensure the scientific authenticity and real-world relevance of our benchmark, we extract 102 tasks from 44 peer-reviewed publications in four disciplines and engage nine subject matter experts to validate them. We unify the target output for every task to a self-contained Python program file and employ an array of evaluation metrics to examine the generated programs, execution results, and costs. Each task goes through multiple rounds of manual validation by annotators and subject matter experts to ensure its annotation quality and scientific plausibility. We also propose two effective strategies to mitigate data contamination concerns. Using our benchmark, we evaluate five open-weight and proprietary LLMs, each with three frameworks: direct prompting, OpenHands CodeAct, and self-debug. Given three attempts for each task, the best-performing agent can only solve 32.4% of the tasks independently and 34.3% with expert-provided knowledge. These results underscore the limited capacities of current language agents in generating code for data-driven discovery, let alone end-to-end automation for scientific research.

## 1 Introduction

Large language models (LLMs) have shown remarkable capabilities beyond text generation, including reasoning (Wei et al., 2022; Yao et al., 2023), tool learning (Schick et al., 2023; Wang et al., 2024a), and code generation (Chen et al., 2021; Yang et al., 2024a). These abilities have piqued significant research interests in developing LLM-based language agents to automate scientific discovery end-to-end. For instance, Majumder et al. (2024a) urge the community to build automated systems for end-to-end *data-driven discovery*, an increasingly important workflow in many disciplines (Hey et al., 2009) that leverages existing datasets to derive new findings. More recently, Lu et al. (2024) claim to have built The AI Scientist, an agent that is capable of automating the entire research workflow, from generating ideas to running experiments and writing papers. This ambitious claim has sparked both excitement and skepticism about the true capabilities of such agents.

---

[*]Correspondence to: {chen.8336,chen.10216,sun.397}@osu.edu

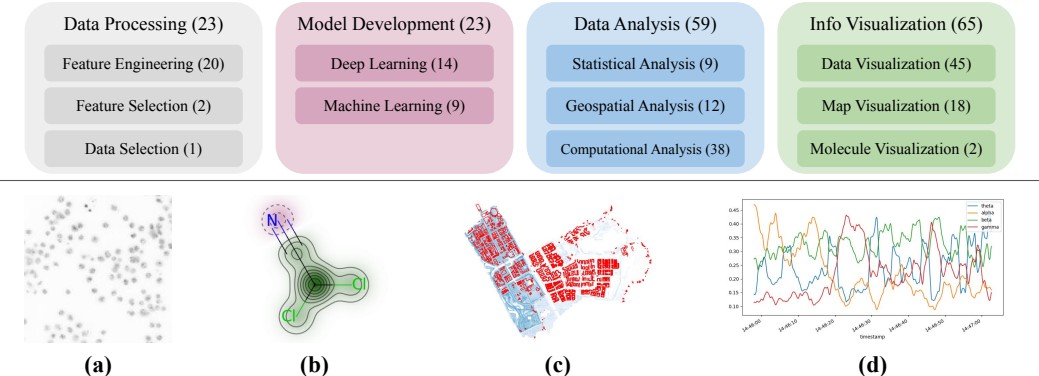

Figure 1: **Top:** Distribution of sub-tasks in ScienceAgentBench. Each task in our benchmark consists of one or more of these sub-tasks and requires successful completion of all sub-tasks to achieve the task goal. **Bottom:** Heterogeneous datasets involved: **(a)** a cell image in Bioinformatics, **(b)** a molecular activity visualization in Computational Chemistry, **(c)** a flooding risk map in Geographical Information Science, and **(d)** an EEG time series in Psychology and Cognitive Neuroscience.

In this work, we contend that for a language agent to fully automate data-driven discovery, it must be able to complete all essential tasks in the workflow, such as model development, data analysis, and visualization. Thus, we advocate careful evaluations of the agents' performance on these tasks, before claiming they can automate data-driven discovery end-to-end. Such an assessment strategy helps grasp a more solid understanding of an agent's strengths and limitations than purely relying on end-to-end evaluations, e.g., using an LLM-based reviewer to assess generated papers (Lu et al., 2024). Yet, high-quality benchmarks focusing on individual tasks in real-world scientific workflows are lacking for objective assessment and continued development of agents for data-driven discovery.

To this end, we present ScienceAgentBench, a new benchmark for evaluating language agents for data-driven discovery. The construction of ScienceAgentBench follows three key design principles. **(1) Scientific authenticity through co-design with subject matter experts**: We ensure the authenticity of tasks in our benchmark by directly extracting them from peer-reviewed publications and engaging nine subject matter experts (incl. senior Ph.D. students and professors) from the respective disciplines to validate them. This approach also minimizes the generalization gap for agents developed on our benchmark to real-world scenarios. In total, we curate 102 diverse tasks from 44 peer-reviewed publications in four disciplines: Bioinformatics, Computational Chemistry, Geographical Information Science, and Psychology & Cognitive Neuroscience (Figure 1). **(2) Rigorous graded evaluation**: Reliable evaluation for language agents is notably difficult due to the open-endedness and complexity of data-driven discovery tasks. We first unify the target output for every task as a self-contained Python program, and then employ an array of evaluation metrics that examine the generated programs, execution results (e.g., rendered figures or test set predictions), and costs. We also provide step-by-step rubrics specific to each task to enable graded evaluation. **(3) Careful multi-stage quality control**: Each task goes through multiple rounds of manual validation by annotators and subject matter experts to ensure its quality and scientific plausibility. We also propose two effective strategies to mitigate data contamination concerns due to LLM pre-training.

We comprehensively evaluate five open-weight and proprietary LLMs, each with three frameworks: direct prompting, OpenHands CodeAct (Wang et al., 2024c), and self-debug. Surprisingly, without expert-provided knowledge, Claude-3.5-Sonnet using self-debug can successfully solve **10.8%** more tasks than using OpenHands CodeAct while costing **17** times less API fees. This result resonates with recent findings that agent designs should jointly consider costs and performance to maximize their practical utility (Kapoor et al., 2024). Still, given three attempts for each task, the best agent can only solve 32.4% of the tasks independently and 34.3% of them with expert-provided knowledge. These results also suggest language agents cannot yet automate essential tasks in data-driven discovery nor the research pipelines end-to-end, in contrast to claims in recent work such as Lu et al. (2024).

Despite their current mediocre performance, we believe language agents hold significant potential in augmenting human scientists' productivity: For each task in our benchmark, it takes a trained

Train a multitask model on the Clintox dataset to predict a drug's toxicity and FDA approval status. Save the test set predictions, including the SMILES representation of drugs and the probability of positive labels, to "pred_results/clintox_test_pred.csv".

**(a) Task Instruction**

1. *On the task*: The ClinTox dataset contains drugs approved by ……
2. *On featurization*: To represent the molecular structure, use Extended-Connectivity Fingerprints (ECFPs) featurization in deepchem……

**(c) Expert-Provided Knowledge**

*Dataset Directory*:
```
|-- clintox/
|---- clintox_test.csv
|---- clintox_train.csv
```

*Dataset Preview*:
[START Preview of clintox/clintox_train.csv]
smiles,FDA_APPROVED,CT_TOX
CCC(/C=C/Cl)(C#C)O,1,0
C[C@H]1C[C@H]2[C@@H]3CC[C@@H]([C@]3(C[C@@H]([C@@H]2[C@@]4(C1=CC(=O)CC4)C)O)C)C(=O)C,1,0
C[C@@H]1CCN([C@H](C1)C(=O)[O-])C(=O)[C@@H](CCC[NH+]=C(N)N)NS(=O)(=O)c2cccc3c2NC[C@@H](C3)C,1,0
...
[END Preview of clintox/clintox_train.csv]

**(b) Dataset Information**

```python
import deepchem as dc
……
from deepchem.molnet.load_function.molnet_loader  import _MolnetLoader

class MyClintoxLoader(_MolnetLoader):
    def create_dataset(self):
        ……

CLINTOX_TASKS = ['FDA_APPROVED', 'CT_TOX']
train_loader = MyClintoxLoader('ECFP', ……)
train_dataset = ……
test_loader = MyClintoxLoader('ECFP', ……)
test_dataset = ……
……
model = dc.models.MultitaskClassifier(……)
model.fit(train_dataset)
test_scores = model.predict(test_dataset, ……)
……
test_scores_df.to_csv('pred_results/clintox_test_pred.csv')
```

**(d) Annotated Program**

Figure 2: An example Computational Chemistry task in ScienceAgentBench with four components.

annotator at least 2.5–3 hours on average to adapt an existing program from public sources, and potentially much longer for a subject matter scientist to write the program from scratch. In contrast, a language agent can usually generate a meaningful program draft within 10 minutes. In the long run, ScienceAgentBench will serve as a benchmark for rigorously measuring progress toward developing language agents to assist scientists in data-driven scientific discovery.

## 2 SCIENCEAGENTBENCH

In this section, we introduce ScienceAgentBench, which aims to evaluate agents on essential tasks in a data-driven discovery workflow. Before automating the entire workflow end-to-end, we envision language agents to first serve as *science co-pilots* that can write code to process, analyze, and visualize data. Similar to co-pilots for software development, we target scientist users who might know how to write such code but want to save hours of programming effort with language agents. Hence, we formulate each task as a code generation problem, whose output is easily verifiable and directly usable by a scientist without additional modification efforts.

### 2.1 PROBLEM FORMULATION

Given a natural language instruction, a dataset, and some optional expert-provided knowledge, an agent shall generate a program to complete the assigned task and save it to Python source code file. Each instance in our benchmark contains four components (Figure 2):

**(a) Task Instruction**, which describes the goal of an essential task in data-driven discovery and its output requirements. To resemble real-world settings, we keep the instructions concise and avoid unnecessary details when describing task goals. This setup also retains the open-endedness of data-driven discovery and encourages the development of practical agents that do not rely on prescriptive directions from scientists. We provide example task instructions in Appendix C for each discipline.

**(b) Dataset Information**, which contains the dataset's directory structure and a preview of its content. For agents without file navigation tools, they need such information to correctly use the dataset in their generated programs. For agents that can navigate file systems, it also helps them save a few turns of interactions to read datasets from the programming environment.

**(c) Expert-Provided Knowledge**, which includes explanations for scientific terms, formulas to conduct analysis, and example usages of programming tools. These pieces of knowledge are provided by subject matter experts, including senior Ph.D. students and professors, and are optional inputs to an agent. In Section 4, we show that while with such information, language agents' knowledge gap in involved disciplines can be mitigated to some extent, they still fall short utilizing it effectively.

**(d) Annotated Program**, which is adapted from an open-source code repository released by a peer-reviewed scientific publication. As shown in Figure 2, each program is self-contained with package imports, function and class implementations, and a main procedure to carry out the task. An agent is expected to produce similar programs that can be executed independently, e.g. by a Python interpreter, but not necessarily using the same tools as those in the annotated programs.

## 2.2 DATA COLLECTION

**Task Annotation.** We start by forming a group of nine graduate students to annotate the tasks in four disciplines: Bioinformatics, Computational Chemistry, Geographical Information Science, and Psychology & Cognitive Neuroscience. Within each discipline, we search for peer-reviewed publications that release their code and data under permissive licenses (Appendix J). Then, we follow five steps to annotate each task: **(1)** Identify a reasonably documented code example that is self-contained and convert it into a task in our benchmark. **(2)** Collect and preprocess datasets used in the code. **(3)** Annotate the reference program by revising the referred code to analyze datasets in our benchmark. **(4)** Implement task-specific success criteria as an executable script and use GPT-4o to draft fine-grained rubrics for evaluation. **(5)** Write the instruction and dataset information for this task. We gathered 110 tasks initially but discarded four because their programs require long execution time or nontrival environment setup. This leaves us with 106 tasks for validation.

**Data Contamination and Shortcut Mitigation.** In our preliminary studies, we have noticed that some agents, such as OpenHands, may take shortcuts to solve a task. For example, when asked to develop a machine learning model, they may directly read and report the ground-truth labels in the test set without writing the training code. Such perfect results are actually cheating and will hurt evaluation validity. In addition, because datasets and programs in our benchmark are open-sourced, they are subject to data contamination in LLM training. To mitigate these issues, we devise two strategies to modify the datasets: **(1)** For each dataset, we randomly remove five data points from its test set. If an LLM-generated program uses automatic data loaders that appeared in the training corpora, it will produce results misaligned to our setup and fail the success criteria. In some cases, we have to skip this step if it would break the completeness of a dataset, e.g., if it results in an incomplete geographical map. **(2)** For tasks involving model development, we re-split the dataset, keep the test set labels only for evaluation, and replace them with dummy values, such as $-1$ for classification tasks. These two strategies effectively mitigate data contamination and agent shortcut concerns by failing agents that recite memorized code or attempt to directly report test set labels. See Appendix F.2: Example F.4 for a case study.

**Expert Validation.** We engage nine subject matter experts, including senior Ph.D. students and professors from the four involved disciplines, to validate each task and provide additional knowledge. For each task, we present to experts with its instruction, dataset information, annotated program, and task rubrics. The experts are asked to validate the tasks by completing a questionnaire (Appendix G), which can be summarized as four steps: **(1)** Validate if an annotated task represents a realistic task in their data-driven discovery workflow. **(2)** Review whether a task instruction gives an accurate high-level description of the program and uses professional languages in their disciplines. **(3)** Provide up to three pieces of knowledge that might be needed for solving each task. **(4)** Make necessary revisions to the rubrics for grading the program. Then, following the experts' feedback, we revise 41 task instructions and remove three tasks that are not representative enough for scientific workflows in their disciplines. With 103 tasks remaining, our publication-oriented annotation strategy is shown to be effective in collecting real-world tasks.

**Annotator Verification.** To ensure data quality, we work with the nine annotators for another round of task verification. We ask the annotators to verify tasks that are not composed by themselves and execute programs to reproduce the results. During this process, we refine 29 task annotations and discard one more task whose result is hard to replicate with the same program due to randomness. We finalize ScienceAgentBench with 102 high-quality tasks for data-driven scientific discovery.

## 2.3 EVALUATION

While it is a preferable feature, the open-endedness of tasks in our benchmark introduces a crucial evaluation challenge. Specifically, our evaluation strategy has to accommodate diverse setup requirements of programs generated by different agents. To address this challenge, we implement

Table 1: Representative examples of task-specific success criteria in ScienceAgentBench. To keep the table concise, we omit output requirements in the task instructions and show the task goals. We provide more details about how these criteria are established in Appendix D.2.

| Task Instruction | Subtasks | Success Criteria |
|---|---|---|
| *Train a multitask model on the Clintox dataset to predict a drug's toxicity and FDA approval status.* | Feature Engineering Deep Learning | The trained model gets $\geq 0.77$ ROC-AUC score on the test set. |
| *Develop a drug-target interaction model with the DAVIS dataset to repurpose the antiviral drugs for COVID.* | Feature Engineering Deep Learning | The top-5 repurposed drugs match the gold top-5 drugs. |
| *Analyze the inertial measurement unit (IMU) data collected during sleep and compute sleep endpoints: time of falling asleep, time of awakening, and total duration spent sleeping.* | Computational Analysis | Each computed endpoint is close (`math.isclose` in Python) to the corresponding gold answer. |
| *Analyze Toronto fire stations and their service coverage. Visualize the results to identify coverage gaps.* | Map Visualization | The resulting figure gets $\geq 60$ score by the GPT-4o Judge. |

a pipeline to set up a conda environment flexibly for any program. Before evaluation, the conda environment is initialized with seven basic Python packages: numpy, pandas, matplotlib, pytorch, tensorflow, rdkit, and tf_keras. To evaluate each program, we first use `pipreqs`[1] to analyze it and generate a file listing all packages used. Then, according to the file, we use `pip-tools`[2] and hand-crafted rules to update the conda environment and properly configure the packages. We execute each program in the customized environment and calculate the evaluation metrics.

**Program Evaluation.** We comprehensively evaluate each generated program with four metrics. **(1) Valid Execution Rate (VER)** checks if the program can execute without errors and save its output with the correct file name. **(2) Success Rate (SR)** examines whether a program output meets the success criteria for each task goal (Table 1), such as test set performance, prediction-answer matches, and visualization quality. To automatically check these criteria, we implement them as evaluation programs for each task during annotation. By nature, **SR** is conditioned on valid execution: If a program has execution errors or does not save its output correctly, its **SR** will be 0. Both **VER** and **SR** are binary metrics. **(3) CodeBERTScore (CBS)** measures how closely the generated program resembles the annotated one with contextual embeddings and calculates the F1 metric for matched token embeddings (Zhou et al., 2023). If **SR** = 1 for a program, we change its **CBS** to 1.0 as well to reflect task success. **(4) API Cost (Cost)** calculates the average cost (in USD) to complete one task in our benchmark, since it is important for language agents to control their cost and optimize their design for better practical utility (Kapoor et al., 2024).

**Figure Evaluation.** If the task output is a figure, we follow existing work (Wu et al., 2024; Yang et al., 2024b) to evaluate its quality using GPT-4o as a judge, which is shown to correlate reasonably well with human raters. We use Yang et al. (2024b)'s prompt to request GPT-4o to compare the program-produced figure with the ground-truth and respond with a score on its quality. For evaluation stability, we sample 3 responses and use the average score to compute success rates.

**Rubric-Based Evaluation.** Outcome-based evaluation metrics, which require a program to correctly implement all steps for the task, can sometimes be too stringent. For example, an agent would be underrated by these metrics if it gets all steps right but output formatting wrong. As a complement to the outcome-based metrics, we introduce rubric-based evaluation to assess the generated programs at more fine-grained levels. Considering the characteristics of data-driven discovery tasks, we structure the rubrics into five stages: *Data Loading*, *Data Processing*, *Modeling or Visualization*, *Output formatting*, and *Output Saving*. To accelerate the annotation process, we first use GPT-4o to generate the rubrics by designating multiple milestones with scores for the five stages. Then, each rubric is refined by an expert (Appendix H). In this work, we leverage the rubrics to conduct

---

[1]`https://github.com/bndr/pipreqs`
[2]`https://github.com/jazzband/pip-tools`

Table 2: Comparison of ScienceAgentBench to representative benchmarks. [†] DiscoveryBench-Real is evaluating the quality of generated programs indirectly through the natural language hypothesis, while ScienceAgentBench's focus is to rigorously assess the programs and their execution results.

| Benchmark | Code Gen Complexity | Task Sources | Heterogeneous Data Processing | Shortcut Prevention | Scientific Subjects | # Test Tasks |
|---|---|---|---|---|---|---|
| TaskBench (Shen et al., 2024) | No Code Gen | Synthetic | ✗ | ✗ | 0 | 28,271 |
| SWE-Bench (Jimenez et al., 2024) | File-Level Edit | GitHub | ✗ | ✗ | 1 | 2,294 |
| BioCoder-Py (Tang et al., 2024c) | Function-Level | GitHub | ✗ | ✗ | 1 | 1,126 |
| ML-Bench (Tang et al., 2024b) | Line-Level | GitHub | ✓ | ✗ | 1 | 260 |
| MLAgentBench (Huang et al., 2024b) | File-Level Edit | Kaggle | ✗ | ✗ | 1 | 13 |
| DiscoveryBench-Real (Majumder et al., 2024b) | Code Gen[†] | 27 Publications | ✓ | ✗ | 6 | 239 |
| SciCode (Tian et al., 2024) | Function-Level | Publications | ✗ | ✓ | 5 | 80 |
| BLADE (Gu et al., 2024) | Function-Level | 31 Publications | ✗ | ✗ | 6 | 12 |
| ScienceAgentBench (Ours) | File-Level Gen | 44 Publications | ✓ | ✓ | 4 | 102 |

human evaluation for generated programs (Section 4.2). We deem that automating this rubric-based evaluation approach, such as developing an LLM-based judge, is a meaningful future direction.

## 2.4 COMPARISON WITH EXISTING BENCHMARKS

ScienceAgentBench differs from other benchmarks with a unique ensemble of research challenges (Table 2). **(1)** Tasks in our benchmark require an agent to generate a standalone *program file from scratch*, in contrast to JSON API calls in TaskBench, abstract workflow descriptions in Discovery-Bench, or a few lines of code completion or edits in other benchmarks. To do so, an agent needs to have a deep understanding of the task, decompose it into classes and functions appropriately, and implement them. **(2)** Our benchmark adapts *44 peer-reviewed publications* and covers a variety of real-world datasets in four different disciplines. Compared to ML-Bench and DiscoveryBench, our ScienceAgentBench includes more *heterogeneous datasets* that have complex structures (Figure 1), such as cell images, chemical structure-activity relationships, and geographical maps with multiple layers. **(3)** ScienceAgentBench is also one of the two benchmarks that tries to *mitigate data contamination and agent shortcut issues*, which helps establish valid evaluation. **(4)** Our benchmark has a medium scale of 102 tasks. Although smaller than benchmarks with synthetic or easier tasks, this scale is reasonable to evaluate agents, considering the annotation difficulty and evaluation cost.

## 3 EXPERIMENTAL SETUP

We experiment with three open-weight LLMs, Llama-3.1-Instruct-70B, 405B (Dubey et al., 2024), and Mistral-Large-2 (123B) (MistralAI, 2024), and two proprietary LLMs, GPT-4o (OpenAI, 2024) and Claude-3.5-Sonnet (Anthropic, 2024). For all experiments, we use the same hyperparameters, temperature $= 0.2$ and top_p $= 0.95$, and perform 0-shot prompting[3] via the APIs. The prompts are included in Appendix I. We evaluate the LLMs under three different (agent) frameworks:

**Direct Prompting.** Direct prompting is a simple framework that does not interact with any programming environment. Given the task inputs, it prompts an LLM to generate a corresponding program in one pass. We use this framework to show the basic code generation capability of each LLM.

**OpenHands CodeAct.** OpenHands[4] (Wang et al., 2024c) is a generalist agent development framework for code generation and software engineering. It provides three kinds of tools in the environment and defines different actions for an agent to interact with the tools: Python code interpreter, bash shell, and web browser. Among the agents developed with OpenHands, we use the best-performing CodeActAgent v1.9 (Wang et al., 2024b) in our experiments. This agent unifies all actions in OpenHands, including the agent-computer interface commands (Yang et al., 2024a) to read and edit local files, into a large action space of different Python API calls. We experiment with the CodeActAgent v1.9 using different LLMs to test the effectiveness of OpenHands' framework

---

[3]OpenHands has a built-in 1-shot example to demonstrate response formats, tool usages, and other plugins like web browser. We do not provide any examples from our benchmark when evaluating OpenHands.

[4]`https://github.com/All-Hands-AI/OpenHands`, originally named as OpenDevin.

design for code generation tasks in data-driven discovery. For simplicity, we shorten the name of this agent framework as OpenHands CodeAct.

**Self-Debug.** Self-debug (Chen et al., 2024a) is a code generation framework for LLMs to execute their generated programs, access execution results, and then reflect on the results to improve each program iteratively. In this work, we re-implement self-debug with three modifications. First, we do not instruct the LLMs to generate reflections before debugging the code, since self-reflection may not always yield better results (Chen et al., 2024b; Huang et al., 2024a; Jiang et al., 2024). Second, we allow early exits if the backbone LLM generates the same program for two consecutive debugging turns. Finally, before running each program, we use `pipreqs` and `pip-tools` to set up the environment. We do not initialize the self-debug environment with any of the basic packages or provide the rules to configure some packages that are used for evaluation (Section 2.3). Even though self-debug might not be able to use some packages due to this design choice, we want to ensure fair comparisons with other baselines, which also have no access to these information.

To improve evaluation stability, we repeat each task with three independent runs in all experiments. Then we select the best run according to the metrics in the following order: maximum **SR**, maximum **VER**, maximum **CBS**, and minimum **Cost**. We refer to the next metric in this order to break ties. For example, if two programs generated for a task both have **SR** $= 0$, we pick the one with higher **VER**. Finally, we report each metric based on the average performance of selected runs. We also include the mean performances out of three runs and standard deviations in Appendix E.1.

## 4 RESULTS AND ANALYSIS

Through comprehensive experiments (Table 3), we show that the latest LLMs and agents can only achieve low-to-moderate task success rates. Given three attempts for each task, Claude-3.5-Sonnet with self-debug demonstrates the best performance (34.3% **SR**) when using expert-provided knowledge. This result underline that LLM-based agents are not yet capable of fully addressing realistic and challenging data-driven discovery tasks, such as those in ScienceAgentBench.

### 4.1 MAIN RESULTS

**Direct Prompting vs. Self-Debug: Execution feedback is necessary for LLMs to generate useful programs.** As shown in Table 3, directly prompting LLMs cannot unleash their full potential in programming for data-driven discovery tasks. Without executing its code, even the best performing LLM, Claude-3.5-Sonnet, can only solve 16.7% of the tasks independently and 20.6% with additional knowledge. For most failed tasks, we share similar findings with Liang et al. (2024) that LLM-generated programs have correct high-level structures but implementation-level errors, such as missing steps or wrong API usage. Compared to direct prompting, self-debug can nearly *double* Claude-3.5-Sonnet's success rate (16.7 → 32.4; 1.94×) without extra knowledge. With expert-provided knowledge, Claude-3.5-Sonnet using self-debug also shows decent improvement over direct prompting. It achieves 13.7 absolute gains on **SR** (20.6 → 34.3; 1.67×) and 45.1 absolute gains on **VER** (41.2 → 86.3; 2.09×). These results highlight the effectiveness of the simple self-debug framework and the importance of enabling LLMs to execute and revise their code for complex tasks.

**OpenHands CodeAct vs. Self-Debug: Agent designs should consider costs and capabilities of LLMs.** For four of the five LLMs evaluated, self-debug demonstrates better performance than OpenHands CodeAct, with GPT-4o as the only exception (Table 3). By examining the trajectories, we find that GPT-4o is better at leveraging tools in OpenHands than other LLMs. For instance, it is the only LLM that search for more details about the provided knowledge with the web browser. In contrast, other LLMs are still struggling with specialized bash commands in OpenHands to edit programs correctly (Example in Appendix F.1). We hypothesize that GPT-4o may have been trained to better follow instructions for language agents and to better use complex tools like a web browser.

When it comes to self-debug, which has a more straightforward design, GPT-4o loses its advantage and underperforms Mistral-Large-2 and Claude-3.5-Sonnet, both of which are trained for better code generation according to their reports (MistralAI, 2024; Anthropic, 2024). Most surprisingly, without the help of expert-provided knowledge, Claude-3.5-Sonnet using self-debug can successfully solve 10.8% more tasks (21.6 → 32.4 **SR**) than using OpenHands while costing 17 times less API fees ($0.958 → $0.057), which is a critical factor to consider for practical applications. Overall, our

Table 3: Results on ScienceAgentBench. The **best performances** (with and without domain knowledge) for each framework are in bold. The overall best performances for each metric are underlined.

| Models | Without Knowledge | | | | With Knowledge | | | |
|---|---|---|---|---|---|---|---|---|
| | SR | CBS | VER | Cost ↓ | SR | CBS | VER | Cost ↓ |
| *Direct Prompting* | | | | | | | | |
| Llama-3.1-Instruct-70B | 5.9 | 81.5 | 29.4 | 0.001 | 4.9 | 82.1 | 27.5 | 0.001 |
| Llama-3.1-Instruct-405B | 3.9 | 79.4 | 35.3 | 0.010 | 2.9 | 81.3 | 25.5 | 0.011 |
| Mistral-Large-2 (2407) | 13.7 | 83.2 | 47.1 | 0.009 | 16.7 | 84.7 | 39.2 | 0.009 |
| GPT-4o | 11.8 | 82.6 | **52.9** | 0.011 | 10.8 | 83.8 | **41.2** | 0.012 |
| Claude-3.5-Sonnet | **17.7** | **83.6** | 51.0 | 0.017 | **21.6** | **85.4** | **41.2** | 0.017 |
| *OpenHands CodeAct* | | | | | | | | |
| Llama-3.1-Instruct-70B | 6.9 | 63.5 | 30.4 | 0.145 | 2.9 | 65.7 | 25.5 | 0.252 |
| Llama-3.1-Instruct-405B | 5.9 | 65.8 | 52.0 | 0.383 | 8.8 | 71.4 | 58.8 | 0.740 |
| Mistral-Large-2 (2407) | 9.8 | 72.5 | 53.9 | 0.513 | 13.7 | 78.8 | 50.0 | 0.759 |
| GPT-4o | 19.6 | 83.1 | 78.4 | 0.803 | **27.5** | **86.3** | 73.5 | 1.094 |
| Claude-3.5-Sonnet | **21.6** | **83.6** | **87.3** | 0.958 | 24.5 | 85.1 | **88.2** | 0.900 |
| *Self-Debug* | | | | | | | | |
| Llama-3.1-Instruct-70B | 13.7 | 82.7 | 80.4 | 0.007 | 16.7 | 83.4 | 73.5 | 0.008 |
| Llama-3.1-Instruct-405B | 14.7 | 82.9 | 78.4 | 0.047 | 13.7 | 83.6 | 79.4 | 0.055 |
| Mistral-Large-2 (2407) | 23.5 | 85.1 | 83.3 | 0.034 | 27.5 | 86.8 | 78.4 | 0.036 |
| GPT-4o | 22.6 | 84.4 | 83.3 | 0.047 | 23.5 | 85.6 | 71.6 | 0.046 |
| Claude-3.5-Sonnet | **32.4** | **86.4** | **92.2** | 0.057 | **34.3** | **87.1** | **86.3** | 0.061 |

results resonate with recent findings on agent design (Kapoor et al., 2024; Xia et al., 2024): **(1)** LLM-based agents do not always benefit from a large action space with complex tools; and **(2)** both cost and performance should be considered when designing or selecting agent frameworks. We also provide a detailed error analysis in Appendix E.2 and identify future research directions.

**With vs. Without Expert-Provided Knowledge: Expert-provided knowledge does not always lead to metric improvement.** On one hand, we observe that expert-provided knowledge leads to consistent improvements on **SR** and **CBS** for most agents (Table 3). These agents can effectively leverage helpful information in the knowledge, such as API names and some concrete steps in the task, to generate a high-quality program draft that closely resembles the annotated gold program and then use execution feedback to address implementation errors.

On the other hand, we notice that there are performance decreases on **VER** for most agents. These decreases can be attributed to two reasons. **(1)** Expert-provided knowledge specifies some specific tools that are less familiar to the agents. Originally, they would only use basic tools like rdkit and sklearn in their generated programs, which are free of execution errors. With provided knowledge, the agents would use those specified tools to generate programs, which often contain incorrect API usage and hallucinated API calls. **(2)** The agents do not know how to solve some tasks without expert-provided knowledge and would generate some executable but less meaningful programs, e.g., to produce an empty figure. While additional knowledge helps them to produce more concrete modeling or analysis, such programs are error-prone and hard to fix with execution feedback (Appendix F.2). For these reasons, despite decreases in **VER**, we argue that expert-provided knowledge helps agents to generate more useful programs from a scientist user's perspective, as reflected by **SR** and **CBS**, and future AI agents should improve their abilities to better leverage such information.

**Language agents cannot solve complex data-driven discovery tasks yet.** Our further analysis on the best performing agent, Claude-3.5-Sonnet with self-debug and expert-provided knowledge, show that it is not yet capable of addressing complex tasks in data-driven discovery. To estimate the complexity of tasks, we visualize the number of lines in their corresponding gold programs using box plot (Figure 3; **Left**). More than 75% of succeeded tasks lean to the simpler side because their gold programs have less than 58.6 lines, which is the mean length of all gold programs in the benchmark. In other words, language agents still fail on many tasks with complex gold programs.

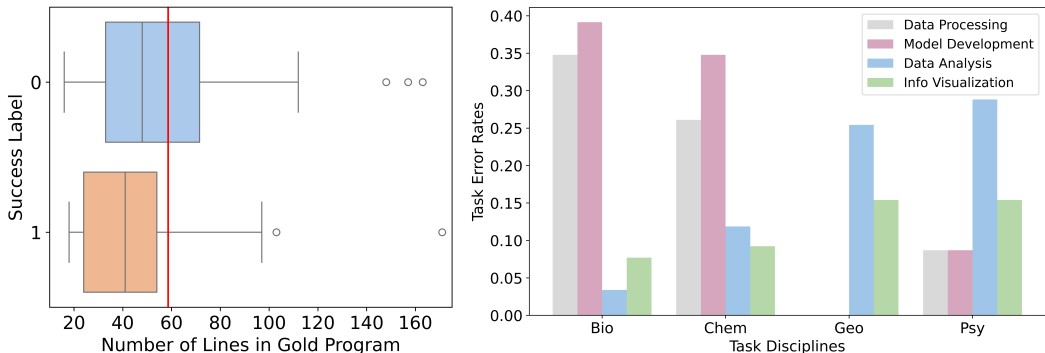

Figure 3: Task performance analysis of Claude-3.5-Sonnet with self-debug and expert-provided knowledge. **Left:** Distribution of lines in gold programs for succeeded and failed tasks. The red vertical line marks the average length (58.6 lines) of all gold programs in the benchmark. **Right:** Task error rates for each sub-task category in each discipline.

To understand the task failures, we break them down by different disciplines and sub-task categories (Figure 3; **Right**). For Bioinformatics and Computational Chemistry, the agent mostly fails on tasks involving data processing and model development. This is because data in these two disciplines are highly heterogeneous, including cell images, molecules, and genes, which can be hard to process. Without correctly processed data, the agent would also not be able to develop and train a functioning model, not to mention choosing appropriate configurations for various models such as Convolutional or Graph Neural Networks used in the tasks. For Geographical Information Science and Psychology & Cognitive Neuroscience, their tasks usually require discipline-specific tools, such as Geopandas and Biopsykit, to analyze the datasets. However, existing LLMs fall short of using these tools and can generate incorrect or hallucinated API usage in the programs. Given these shortcomings, we argue that current language agents cannot yet automate data-driven discovery tasks or the full research pipeline, in contrast to claims made in recent work such as Lu et al. (2024).

## 4.2 HUMAN EVALUATION

**Evaluation Setup.** To further investigate the performance of Claude-3.5-Sonnet with self-debug (the best-performing agent), we conduct a rubric-based human evaluation of all the 102 programs generated using expert-provided knowledge. With the task-specific rubrics validated by experts (examples in Appendix H) and gold programs as references, each generated program is rated by two different evaluators who participated in data collection. To reduce possible noises in ratings, the evaluators only mark whether a rubric item is met by the LLM-generated program. For each stage, we add up points for satisfied rubric items and normalize them by total available points to the range of 0–100. Similarly, we calculate the overall score considering all items. The final score of each program is the average of two evaluators' ratings.

Additionally, one purpose of this human evaluation is to assign partial credits to the generated program even if it is not correct (Section 2.3). Therefore, we do not provide the evaluators with program execution results and hide task success outcomes. Although this setup encourages evaluators to examine LLM-generated programs carefully, it also introduces some noise. For example, there are tasks where both a feed-forward neural network and a random forest model can achieve satisfying performance on the test set. While the gold program implements the neural network, the agent chooses to use random forest. Since each rubric is derived from a gold program and reflect its implementation, there are chances that the evaluator overlooks such equivalence. Also, for output formatting, we observe some subjective variance when judging the formats of figures, such as colors, scales, and text labels, according to the rubrics and gold programs. As a result, successful programs would not always receive a perfect human rating.

**Results and Analysis.** As shown in Figure 4, data loading and processing, the first two stages in data-driven discovery tasks, can distinguish successful programs from failed ones. Except for a few outliers, almost all successful programs receive a perfect human rating for data loading. In contrast, 25% of the failed programs have their rating below 50 in the first stage. For data processing, the rating distribution of successful programs skews toward the full score, while that of failed programs

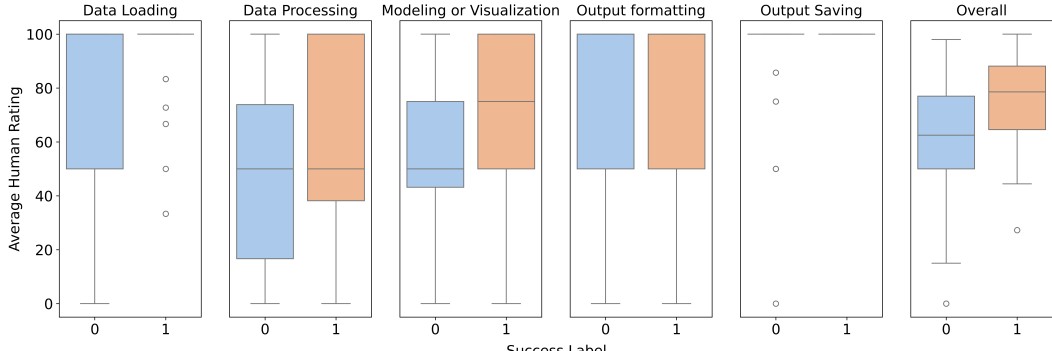

Figure 4: Rubric-based human ratings for 102 programs generated by Claude-3.5-Sonnet with self-debug and expert-provided knowledge. We show the overall distributions and those for the five stages in our rubrics (Section 2.3). The blue boxes are distributions for failed tasks, and the orange ones are for succeeded tasks. The open dots represent outliers in the distributions.

skews toward a score between 20 and 50. These human evaluation results correspond to an intuitive explanation: If the dataset were not loaded or processed correctly, it would be impossible to solve a task successfully, regardless of the code implementation for consequent stages.

In the third stage, modeling or visualization, human ratings for successful and failed programs are also different: The median score of successful programs is already at the 75th percentile of failed program ratings. This indicates that human evaluators agree with the **SR** metric and prefer programs passing all success criteria for the task, even though they may have some minor issues. For output formatting and saving, we find no difference between the two groups of programs, indicating that LLMs like Claude-3.5-Sonnet can follow such instructions reasonably well.

Overall, human ratings for succeeded and failed programs form two overlapped but distinguishable distributions, which meets our motivation to complement outcome-based metrics with fine-grained evaluation. These ratings agree with our main result and suggest that some LLM-generated programs are close to success but hindered by some bottlenecks, such as data loading and processing. Future research may, for example, improve language agents' capability to better process scientific data.

## 5    CONCLUSION

We introduce ScienceAgentBench, a new benchmark to evaluate language agents for data-driven scientific discovery. We compile 102 diverse, real-world tasks from 44 peer-reviewed publications across four scientific disciplines and engage nine subject matter experts to ensure data quality. Through comprehensive experiments on five LLMs and three frameworks, we show that the best-performing agent, Claude-3.5-Sonnet with self-debug, can only solve 34.3% of the tasks when using expert-provided knowledge. Our results and analysis suggest that current language agents cannot yet automate tasks for data-driven discovery or a whole research pipeline. By introducing ScienceAgentBench, we advocate the use of language agents to assist human scientists with tedious tasks in their workflows and call for more rigorous assessments of such agents.

### ETHICS STATEMENT

Our benchmark is constructed by adapting open-source code and data, to which we respect their creators' ownership and intellectual property. In Appendix J, we have made our best effort to cite the original papers, list the repositories, and provide their licenses. Still, we acknowledge that two repositories are copyrighted and believe their terms for use are compatible with our research purpose (Table J.4, J.5). We welcome requests from the original authors to modify or remove relevant tasks if needed.

Meanwhile, agents developed with ScienceAgentBench should consider potential safety issues in deployment, especially when performing Bioinformatics and Computational Chemistry tasks. This work contributes an evaluation benchmark to assess existing language agents rigorously, which has

limited or no risk in inadvertently synthesizing toxic or dangerous chemicals. Yet, we are aware that the safety of language agents for science is an important research topic (Tang et al., 2024a) and have discussed with our subject matter experts about the risk of synthesizing toxic or dangerous chemicals: (1) Our Bioinformatics and Computational Chemistry tasks focus on property prediction, feature analyses, and molecule visualization, which does not involve synthesis or generation of biological or chemical substances. (2) Unlike Coscientist (Boiko et al., 2023), agents evaluated in our submission are not connected to any laboratory hardwares. Thus, it is impossible for these agents to produce any dangerous chemicals or substances on their own. Even if they were to be instructed to write code for chemical synthesis in real-world applications, human intervention is still required to grant the access to laboratories, reagents, and equipment. (3) The target outputs for every task in ScienceAgentBench are unified as self-contained Python programs. Therefore, the evaluated agents only generate code for processing, analyzing and visualizing scientific data that is already publicly available. They are not instructed to generate chemical reactions or synthesis pathways. We also recommend the developers of these agents to consider such potential risks seriously and provide effective intervention and feedback mechanisms for users.

## AUTHOR CONTRIBUTIONS

Z. Chen led the project, formulated the benchmark, organized data collection and human evaluation, implemented programs and evaluation scripts for experiments, conducted all experiments on GPT-4o/Claude/Mistral, and wrote the manuscript. S. Chen designed and implemented rubric-based evaluation, and helped to run direct prompting, self-debug, and OpenHands experiments on Llama 3.1 70B/405B, optimize the evaluation scripts, and revise the manuscript. Y. Ning helped to run some experiments on OpenHands with Llama 3.1 70B/405B. Z. Chen, S. Chen, Y. Ning, Q. Zhang, B. Wang, B. Yu, Y. Li, Z. Liao, and C. Wei worked as the student annotators to collect the benchmark, verify the tasks, and evaluate generated programs based on rubrics. Z. Lu, V. Dey, M. Xue, F. Baker, B. Burns, D. Adu-Ampratwum, X. Huang, X. Ning, and S. Gao are subject matter experts who validated the tasks and provided task-specific knowledge. In addition, S. Gao provided substantial guidance in Geographical Information Science task collection and constructive feedback on the project during weekly discussions. Y. Su and H. Sun are senior authors who oversaw this project, contributed to the core ideas, and revised the manuscript. H. Sun conceived the research problem.

## ACKNOWLEDGMENTS

The authors would thank colleagues from the OSU NLP group and the NSF-funded ICICLE AI Institute for constructive feedback. This research was sponsored in part by NSF OAC 2112606, Amazon, and Ohio Supercomputer Center (Center, 1987). The views and conclusions contained herein are those of the authors and should not be interpreted as representing the official policies, either expressed or implied, of the U.S. government. The U.S. Government is authorized to reproduce and distribute reprints for Government purposes notwithstanding any copyright notice herein.

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

APPENDICES

We provide more details omitted in the main text as follows:

- Appendix A: Limitations and Future Directions

- Appendix B: Related Work

- Appendix C: Example Task Instructions
    - Table C.1: Example Instructions for Bioinformatics and Computational Chemistry Tasks
    - Table C.2: Example Instructions for Geographical Information Science and Psychology & Cognitive Neuroscience Tasks

- Appendix D: More Details about Benchmark Construction
    - Appendix D.1: Details about Annotated Programs
    - Appendix D.2: Details about Success Criteria

- Appendix E: More Details about Main Results
    - Appendix E.1: Mean and Standard Deviations of Agent Performance
    - Appendix E.2: Error Analysis of OpenHands CodeAct and Self-Debug

- Appendix F: Case Studies
    - Appendix F.1: Action Space of OpenHands
    - Appendix F.2: Case 2: Influence of Expert-Provided Knowledge

- Appendix G: Expert Validation Details
    - Appendix G.1: Questionnaire for Domain Experts
    - Appendix G.2: Program Example for Domain Experts
    - Appendix G.3: Knowledge Example for Domain Experts

- Appendix H: Rubric Examples
    - Appendix H.1: An example rubric of a Computational Chemistry task generated by GPT-4o without expert revision
    - Appendix H.2: An example rubric revised by an expert by adding the available points to two items
    - Appendix H.3: An example rubric of a Geographical Information Science task generated by GPT-4o without expert revision
    - Appendix H.4: An example rubric of a Geographical Information Science task revised by an expert by reducing the available points for several items

- Appendix I: Prompt Templates
    - Table I.1: Prompt Template for Direct Prompting
    - Table I.2: Prompt Template for Self-Debug
    - Table I.3: Prompt Template for OpenHands CodeAct

- Appendix J: Publications, Repositories, and Licenses
    - Table J.1: List of Bioinformatics and Computational Chemistry Publications
    - Table J.2: List of Geographical Information Science and Psychology & Cognitive Neuroscience Publications
    - Table J.3: List of Repositories and Licenses
    - Table J.4: Copyright Information for rasterio/rasterio
    - Table J.5: Copyright Information for ackingmaterials/matminer.

## A    LIMITATIONS AND FUTURE DIRECTIONS

**Capabilities and Evaluation of Language Agents for Science.** In this work, we have developed a benchmark focusing on tasks in data-driven discovery and formulate them as code generation problems due to two reasons. (1) Data-driven discovery is an increasingly important workflow for science (Hey et al., 2009). While plenty of computational tools (Cao, 2017) and AI models (Wang et al., 2023) have been developed, the sheer amount and heterogeneity of data are already overwhelming for scientists (Bell et al., 2009), not to mention the programming efforts to access these tools and models for processing, analyzing, and visualizing scientific data. A language agent that can automate such tedious tasks in data-driven discovery would help to save hours of effort for scientists. (2) We aim to rigorously assess the capabilities of existing language agents as science co-pilots that can write code to process, analyze, and visualize data. Hence, we formulate each task as a code generation problem, whose output shall be easily verifiable using well-established automatic metrics and directly usable by a scientist without additional efforts to modify or implement.

As a result, we only focus on the code generation capability of language agents. We encourage future studies to carefully examine the agents' other capabilities that can help with scientific discovery, such as summarizing literature (Lin et al., 2024), suggesting ideas (Si et al., 2024), or planning experiments (Boiko et al., 2023). Specifically, we advocate rigorous, comprehensive assessments of one such capability at a time, as we need to deeply understand the strengths and limitations of current language agents for each aspect of scientific discovery. In addition, while we only use well-established evaluation methods in our benchmark, such as CodeBERTScore (Zhou et al., 2023) and GPT-4o judge for figures (Wu et al., 2024; Yang et al., 2024b), we acknowledge that they are not perfect yet. Future research may leverage the diverse set of tasks in our benchmark to develop better automatic evaluation metrics or human evaluation protocols for data-driven discovery tasks and code generation problems.

**Diversity of Tasks, disciplines, and Programs.** Although we strive to include a diverse set of tasks and programs from different scientific disciplines in ScienceAgentBench, we devise several compromises to make data collection more practical. First, when collecting publications, we have indeed found more with programs written in R, Stata, or Matlab. However, because our annotators are not familiar with these programming languages, we focus on collecting Python programs, which all annotators can adapt confidently. Second, for evaluation efficiency, we only collect programs that can accomplish the task within 10 minutes. As a result, the final benchmark includes relatively fewer tasks that process large-scale data and develop complex methods. Finally, we choose the four representative disciplines considering their abundance of open-source data and the availability of experts we can easily contact. With these limitations in mind, we have designed a principled, extensible data collection process and expert validation protocol. Future work is encouraged to expand ScienceAgentBench with programs in other languages and tasks in other disciplines. We also plan to continually expand our benchmark into more disciplines and facilitate future research in two ways: (1) ScienceAgentBench will serve as a necessary testbed for developing future language agents with stronger capabilities to process scientific data or to utilize expert-provided knowledge. (2) ScienceAgentBench will help future research to design new automatic graded metrics, such as an LLM judge based on task-specific rubrics, to assess language agents for data-driven discovery.

## B  RELATED WORK

**AI for Science.** Since deep learning unlocks the power of data, AI algorithms and models have been increasingly used to accelerate scientific discovery (Wang et al., 2023). One of the most prominent examples is AlphaFold (Jumper et al., 2021), which can predict protein structures with high accuracy and save biologists months to years of effort. More recently, a tremendous number of language models has been developed for different disciplines, including math (Yue et al., 2024), chemistry (Yu et al., 2024), biology (Labrak et al., 2024), geography (Li et al., 2023), and so on. [5] To automate data-driven discovery end-to-end, it is necessary for language agents to write code to access these AI models and other computational tools (Cao, 2017). Our work aims to develop language agents with this essential ability, which can help scientists save hours of programming effort, and rigorously evaluate such agents to grasp a more solid understanding of their strengths and limitations.

**Agents and Benchmarks for Task Automation.** Developing agents for task automation is a long-established challenge in AI research (Russell & Norvig, 2010). Built upon LLMs, a new generation of agents has shown new promise to automatically perform many tasks in web navigation (Deng et al., 2023; He et al., 2024; Koh et al., 2024; Zheng et al., 2024; Zhou et al., 2024), software development (Jimenez et al., 2024; Wang et al., 2024c; Yang et al., 2024a), or scientific discovery (Boiko et al., 2023; Zheng et al., 2023; Lu et al., 2024).

To evaluate the performance of these new agents, many benchmarks have been recently proposed. For example, TaskBench (Shen et al., 2024) is one of the first benchmarks for evaluating language agents with large-scale synthetic tasks. The output of these tasks are formatted as JSON API tool calls, which hinders the generalization of agents developed on this benchmark to constantly changing tools or new ones in practice. To resemble real-world use cases with more flexibility, many benchmarks formulate their tasks as code generation and unifies their outputs as Python programs, such as SWE-Bench (Jimenez et al., 2024), BioCoder-Py (Tang et al., 2024c), ML-Bench (Tang et al., 2024b), and MLAgentBench (Huang et al., 2024b). Yet, these benchmarks only consist of tasks found on secondary sources, such as GitHub and Kaggle.

To fill the gap in evaluating language agents for scientific tasks, a few benchmarks start to use scientific publications as their task sources, including SciCode (Tian et al., 2024), BLADE (Gu et al., 2024), and DiscoveryBench-Real (Majumder et al., 2024b). Among them, DiscoveryBench-Real is the most similar to our ScienceAgentBench. However, DiscoveryBench-Real asks the agents to output abstract steps to complete a task in natural language, which is hard to evaluate rigorously and maybe practically less useful. With ScienceAgentBench, we advocate careful evaluations of language agents' performance on individual tasks, instead of purely relying on end-to-end evaluations of these agents, e.g., using an LLM-based reviewer to assess generated papers (Lu et al., 2024). In the long run, ScienceAgentBench will serve as a high-quality benchmark focusing on essential tasks that involve code generation in real-world data-driven discovery workflows for objective assessment and continued development of future language agents.

---

[5] We refer to Zhang et al. (2024) for a comprehensive survey on scientific language models.

# C  EXAMPLE TASK INSTRUCTIONS

Table C.1: Example instructions of Bioinfomatics and Compuational Chemistry tasks (Section 2.2).

| Domain | Task Instruction |
|---|---|
| Bioinformatics | Train a cell counting model on the BBBC002 datasets containing Drosophila KC167 cells. Save the test set predictions as a single column "count" to "pred_results/cell-count_pred.csv". |
| | Train a drug-target interaction model using the DAVIS dataset to determine the binding affinity between several drugs and targets. Then use the trained model to predict the binding affinities between antiviral drugs and COVID-19 target. Rank the antiviral drugs based on their predicted affinities and save the ordered list of drugs to "pred_results/davis_dti_repurposing.txt", with one SMILES per line. |
| | Plot the Tanimoto similarities of the fingerprint between the frames. Specifically, the interaction fingerprints between a selected ligand and protein for the first 10 trajectory frames. Save the png file into pred_results/ligand_similarity_pred.png. |
| | Train a VAE model on the given data and perform a 1-vs-all differential expression test for each cell type. Extract top markers for each cell type using the results. Visualize them as a dotplot with the cell types organized using a dendrogram. Save the figure to pred_results/hca_cell_type_de.png. |
| Computational Chemistry | Train a multitask model on the Clintox dataset to predict a drug's toxicity and FDA approval status. Save the test set predictions, including the SMILES representation of drugs and the probability of positive labels, to "pred_results/clintox_test_pred.csv". |
| | Generate features for the given diffusion data based on material composition and use the SHAP feature selection approach to select 20 features. Save the selected features as a CSV file "mat_diffusion_features.csv" to the folder "pred_results/". |
| | Filter the compounds in "hits.csv" and save the SMILES representations of the left ones. Compounds to be kept should have no PAINS or Brenk filter substructures and have a maximum tanimoto similarity of less than 0.5 to any of the active compounds in "train.csv". Save the SMILES of left compounds to "pred_results/compound_filter_results.txt", with each one in a line. |
| | Train a graph convolutional network on the given dataset to predict the aquatic toxicity of compounds. Use the resulting model to compute and visualize the atomic contributions to molecular activity of the given test example compound. Save the figure as "pred_results/aquatic_toxicity_qsar_vis.png". |

Table C.2: Example instructions of Geographical Information Science and Psychology & Cognitive Neuroscience tasks (Section 2.2).

| Domain | Task Instruction |
|---|---|
| Geo Information Science | Analyze and visualize Elk movements in the given dataset. Estimate home ranges and assess habitat preferences using spatial analysis techniques. Identify the spatial clusters of Elk movements. Document the findings with maps and visualizations. Save the figure as "pred_results/Elk_Analysis.png". |
| | Analyze the impact of land subsidence on flooding based on future elevation data of the study area. Identify flood-prone areas and estimate potential building damage to support urban planning and mitigation strategies. Save the results to "pred_results/flooding_analysis.png". |
| | Calculate the deforestation area percentage in the Brazilian state of Rondônia within the buffer zone of 5.5km around road layers. Save the percentage result in a CSV file named "pred_results/deforestation_rate.csv" with a column title percentage_deforestation. |
| | Load North America climate data in NetCDF file and extract temperature data along the time series, then perform a quadratic polynomial fit analysis on the temperature data, and output the fitting results by year in 'pred_results/polynomial_fit_pred.csv'. |
| Psy & Cognitive Neuroscience | Process and visualize the given ECG data by perform R peak detection and outlier correction. Plot an overview of the data and save the final figure as "pred_results/ecg_processing_vis1_pred_result.png". |
| | Analyze the inertial measurement unit (IMU) data collected during sleep and compute sleep endpoints. Load the given data and compute the following sleep endpoints: time of falling asleep, time of awakening, and total duration spent sleeping. The three values should be saved in a JSON file "pred_results/imu_pred.json", and the keys for them are "sleep_onset", "wake_onset", and "total_sleep_duration", respectively. |
| | Analyze cognitive theories using pattern similarity. Process CSV files containing model predictions for various syllogistic reasoning tasks. Calculate similarity scores between these models and pre-computed high-conscientiousness and high-openness patterns. The results will contain similarity scores for each cognitive model with respect to the personality trait patterns. Save the results to "pred_results/CogSci_pattern_high_sim_data_pred.csv". |
| | Train a linear model to learn the mapping of neural representations in EEG signals from one subject (Sub 01) to another (Sub 03) based on the preprocessed EEG data from Sub 01 and Sub 03. Then use the test set of Subject 1 (Sub 01) to generate EEG signal of Subject 3 (Sub 03). Save the generated EEG signal of Subject 3 to "pred_results/linear_sub01tosub03_pred.npy". |

# D   MORE DETAILS ABOUT BENCHMARK CONSTRUCTION

## D.1   DETAILS ABOUT ANNOTATED PROGRAMS

The annotated program for each task is first extracted as is, instead of written by humans or generated by any models, from the open-source repositories of peer-reviewed publications to ensure their scientific authenticity. Then, our annotators make necessary modifications to remove redundant lines and load the datasets in our benchmark. Finally, the annotated programs are validated by subject matter experts, as well as other annotators.

## D.2   DETAILS ABOUT SUCCESS CRITERIA

The success criteria in our benchmark are tailored to each task and established by measuring whether an LLM-generated program accurately reproduces the result of the annotated program. Since the annotated programs are adapted from open-source repositories of peer-reviewed publications and validated by subject matter experts, their execution results faithfully represent part of the research outcomes in those publications. An agent that is capable of implementing a program correctly to reproduce the result would also produce a correct program for similar tasks in real-world scenarios.

For example, we have executed our annotated program to train a multitask model on the Clintox dataset for five independent runs and consistently observe that the model achieves at least 0.77 ROC-AUC score on the test set. Thus, we use 0.77 as the performance threshold in this success criterion and require the agent to train a model with the same level of performance to be considered successfully completing the task. Evaluation criteria for other tasks are also established following the same principle of reproducing some data-driven discovery results.

# E   MORE DETAILS ABOUT MAIN RESULTS

## E.1   MEAN AND STANDARD DEVIATIONS OF AGENT PERFORMANCE

In the main text, we present our results by selecting the best of three independent runs for each task in all experiments (Section 3). For comprehensiveness, we show the mean performances of each agent and standard deviations below, which demonstrate the same findings as in our main results in Section 4.1.

Table E.1: Mean performances of each agent and standard deviations on ScienceAgentBench **without** domain knowledge.

| Models | SR | CBS | VER | Cost ↓ |
|---|---|---|---|---|
| *Direct Prompting* | | | | |
| Llama-3.1-Instruct-70B | 3.6 (2.0) | 81.0 (0.4) | 22.2 (0.9) | 0.001 (0.000) |
| Llama-3.1-Instruct-405B | 3.6 (0.5) | 79.3 (0.1) | 32.0 (0.5) | 0.011 (0.000) |
| Mistral-Large-2 (2407) | 10.1 (1.2) | 82.5 (0.2) | 36.6 (0.9) | 0.010 (0.000) |
| GPT-4o | 7.5 (0.5) | 81.7 (0.1) | 42.2 (1.6) | 0.011 (0.000) |
| Claude-3.5-Sonnet | 11.8 (2.1) | 82.5 (0.4) | 36.0 (1.2) | 0.017 (0.000) |
| *OpenHands CodeAct* | | | | |
| Llama-3.1-Instruct-70B | 3.3 (0.5) | 59.9 (1.6) | 17.0 (1.2) | 0.234 (0.026) |
| Llama-3.1-Instruct-405B | 2.6 (0.9) | 59.0 (4.9) | 34.3 (9.2) | 0.576 (0.108) |
| Mistral-Large-2 (2407) | 7.5 (0.9) | 70.4 (1.1) | 42.8 (1.7) | 0.735 (0.025) |
| GPT-4o | 13.1 (2.6) | 80.6 (1.2) | 62.8 (2.9) | 1.093 (0.071) |
| Claude-3.5-Sonnet | 14.1 (1.2) | 81.2 (0.8) | 63.4 (6.5) | 1.122 (0.056) |
| *Self-Debug* | | | | |
| Llama-3.1-Instruct-70B | 7.2 (1.2) | 81.2 (0.3) | 67.3 (2.4) | 0.009 (0.000) |
| Llama-3.1-Instruct-405B | 8.8 (1.4) | 80.8 (0.5) | 67.0 (2.8) | 0.054 (0.005) |
| Mistral-Large-2 (2407) | 16.0 (1.7) | 83.2 (0.4) | 70.3 (2.6) | 0.043 (0.001) |
| GPT-4o | 14.7 (3.2) | 82.6 (0.6) | 71.2 (1.2) | 0.057 (0.006) |
| Claude-3.5-Sonnet | 22.9 (2.0) | 84.2 (0.3) | 84.0 (1.2) | 0.066 (0.005) |

Table E.2: Mean performances of each agent and standard deviations on ScienceAgentBench **with** domain knowledge.

| Models | SR | CBS | VER | Cost ↓ |
|---|---|---|---|---|
| *Direct Prompting* | | | | |
| Llama-3.1-Instruct-70B | 2.6 (0.5) | 81.7 (0.1) | 19.3 (1.7) | 0.001 (0.000) |
| Llama-3.1-Instruct-405B | 2.9 (0.0) | 81.3 (0.0) | 24.5 (0.0) | 0.011 (0.000) |
| Mistral-Large-2 (2407) | 11.4 (1.2) | 83.8 (0.2) | 28.8 (2.3) | 0.010 (0.000) |
| GPT-4o | 8.2 (1.8) | 83.2 (0.4) | 35.6 (1.8) | 0.012 (0.000) |
| Claude-3.5-Sonnet | 16.7 (2.4) | 84.5 (0.4) | 33.0 (1.2) | 0.017 (0.000) |
| *OpenHands CodeAct* | | | | |
| Llama-3.1-Instruct-70B | 1.6 (0.9) | 60.5 (0.9) | 16.7 (0.8) | 0.296 (0.003) |
| Llama-3.1-Instruct-405B | 4.3 (2.0) | 62.9 (6.3) | 35.6 (1.7) | 0.653 (0.072) |
| Mistral-Large-2 (2407) | 9.2 (0.9) | 74.1 (2.9) | 35.3 (0.8) | 0.757 (0.049) |
| GPT-4o | 16.7 (2.8) | 83.7 (0.7) | 60.8 (2.4) | 1.402 (0.055) |
| Claude-3.5-Sonnet | 15.7 (2.1) | 82.8 (0.3) | 68.0 (3.3) | 1.095 (0.087) |
| *Self-Debug* | | | | |
| Llama-3.1-Instruct-70B | 9.8 (2.1) | 82.0 (0.4) | 60.8 (2.1) | 0.011 (0.000) |
| Llama-3.1-Instruct-405B | 8.2 (0.9) | 82.2 (0.1) | 61.1 (3.8) | 0.072 (0.002) |
| Mistral-Large-2 (2407) | 18.3 (0.5) | 84.9 (0.1) | 62.8 (0.0) | 0.051 (0.001) |
| GPT-4o | 15.0 (2.0) | 83.8 (0.4) | 61.4 (1.7) | 0.063 (0.001) |
| Claude-3.5-Sonnet | 27.8 (2.0) | 85.5 (0.5) | 81.1 (0.9) | 0.072 (0.005) |

## E.2 ERROR ANALYSIS OF OPENHANDS CODEACT AND SELF-DEBUG

Using Claude-3.5-Sonnet as the base LLM, we sample 50 error trajectories for OpenHands CodeAct and self-debug respectively. From the 100 error trajectories, we find that both agents need **better reasoning and self-verification capabilities** to make sure their executable programs are also semantically correct (29/50 errors for OpenHands CodeAct and 30/50 errors for self-debug). For instance, when having trouble loading the actual scientific data, the agent may write code to simulate some fake data to make the program executable but produce incorrect results. Similarly, when the agent cannot implement something correctly, e.g., a graph convolutional neural network, it may just turn to implementing a simpler feed-forward network, which underfits the complex data and cannot reproduce the desired performance. These executable but functionally incorrect programs need to be better captured and fixed by improving the agents' reasoning and self-verification in future research.

The other major issue for both agents is their ability to **install and configure the environments with domain-specific tools correctly**. Our analysis reveals that both the LLM-generated installation commands in OpenHands CodeAct (10/50 are configuration errors) and human-developed packages used in self-debug (9/50 are configuration errors) are not sufficient to set up some domain-specific tools correctly. This finding echoes with concurrent work (Bogin et al., 2024) that environmental setup for scientific tasks remains challenging for language agents. When the environment is not set up correctly, both agents try to get around domain-specific tools in their programs, such as developing a random forest model with scikit-learn instead of deep learning models in deepchem.

Finally, we find that in 23 of the 50 error trajectories, Claude-3.5-Sonnet was struggling with the specialized commands in OpenHands to edit programs correctly (Section 4.1), especially for longer programs. It would fall into loops of repeatedly generating such commands as shown in the case study in Appendix D.1. Such behaviors waste quite a few turns on fixing the use of these commands and largely increase the API cost. Future agent research should reconsider the use of such commands and compare closely with some pipeline-based approaches (Xia et al., 2024).

# F CASE STUDIES

## F.1 CASE 1: ACTION SPACE OF OPENHANDS

In Section 4, our results show that, for four of the five LLMs evaluated, self-debug demonstrates better performance than OpenHands, with GPT-4o as the only exception. For GPT-4o, which may have been trained to better follow instructions for language agents, we indeed observe that it can better leverage tools in OpenHands, such as the web browser commands (lines 33, 60, and 87 in Listing F.1). In sharp contrast, as shown in Listing F.2, Claude-3.5-Sonnet is still struggling with the specialized bash command in OpenHands to update the program file without duplication (line 11 in Listing F.2), and after five steps of trials, it ends up simply using the Python function `open()` instead to edit the file (line 146 in Listing F.2). Thus, LLM-based agents do not always benefit from a large action space with complex tool usage, such as the design of OpenHands.

Listing F.1: An example of GPT-4o using the web browser in OpenHands. Browsing actions in lines 31–34, 58–61, 85–88.

```
1  {
2    ......(first seven turns omitted),
3    [
4      {
5        "id": 17,
6        "timestamp": "2024-09-19T14:13:52.407648",
7        "source": "agent",
8        "message": "I'm asking BrowsingAgent for help with this task.",
9        "action": "delegate",
10       "args": {
11         "agent": "BrowsingAgent",
12         "inputs": {
13           "task": "It seems that the `rsp_rrv` function does not accept a `peaks`
       argument. Let's check the NeuroKit2 documentation to find the correct usage of
       the `rsp_rrv` function.\n\n### Step 8: Check NeuroKit2 Documentation\nWe will
       browse the NeuroKit2 documentation to find the correct usage of the `rsp_rrv`
       function.. I should start with: Search for \"NeuroKit2 rsp_rrv function
       documentation\""
14         },
15         "thought": ""
16       }
17     },
18     {
19       "message": "No observation",
20       "observation": "null",
21       "content": "",
22       "extras": {}
23     }
24   ],
25   [
26     {
27       "id": 20,
28       "timestamp": "2024-09-19T14:13:53.634697",
29       "source": "agent",
30       "message": "Executing browser actions: goto('https://neurokit2.readthedocs.io
       /')",
31       "action": "browse_interactive",
32       "args": {
33         "browser_actions": "goto('https://neurokit2.readthedocs.io/')",
34         "thought": "In order to accomplish my goal, I need to search for the
       NeuroKit2 `rsp_rrv` function documentation. I will start by navigating to the
       NeuroKit2 documentation website.",
35         "browsergym_send_msg_to_user": ""
36       }
37     },
38     {
39       "id": 21,
40       "timestamp": "2024-09-19T14:13:56.398162",
41       "source": "agent",
42       "message": "Visited https://neurokit2.readthedocs.io/en/legacy_docs/",
43       "cause": 20,
44       "observation": "browse",
45       "content": "<::before> NeuroKit \n\nlegacy_docs \n\n  * Documentation \n\n\n\
       n__ NeuroKit \n\n<::before><::after>\n  * Docs  \u00bb \n  * Documentation \n  *
        [ <::before> Edit on GitHub ](https://github.com/neuropsychology/NeuroKit/blob/
       legacy_docs/docs/index.rst)\n\n\n\n* * *\n\n#  Documentation  <::after>\u00b6 \n
       \n<::before>Warning \n\nThe **NEW DOCUMENTATION** website is at [ https://
       neuropsychology.github.io/NeuroKit ](https://neuropsychology.github.io/NeuroKit
       /)\n\n* * *\n\n\u00a9 Copyright 2020, Dominique Makowski  Revision ` bd105f7f `
```

```
     . \n\n<::before><::after> <::before> Read the Docs  v: legacy_docs  <::before>\n
     \nVersions \n     [ legacy_docs ](https://neurokit2.readthedocs.io/en/
     legacy_docs/)\n\nDownloads \n     [ PDF ](//neurokit2.readthedocs.io/_/downloads
     /en/legacy_docs/pdf/)\n     [ HTML ](//neurokit2.readthedocs.io/_/downloads/en/
     legacy_docs/htmlzip/)\n     [ Epub ](//neurokit2.readthedocs.io/_/downloads/en/
     legacy_docs/epub/)\n\nOn Read the Docs \n     [ Project Home ](//readthedocs.org
     /projects/neurokit2/)\n     [ Builds ](//readthedocs.org/projects/neurokit2/
     builds/)\n     [ Downloads ](//readthedocs.org/projects/neurokit2/downloads/)\n\
     nOn GitHub \n     [ View ](https://github.com/neuropsychology/NeuroKit/blob/
     legacy_docs/docs/index.rst)\n     [ Edit ](https://github.com/neuropsychology/
     NeuroKit/edit/legacy_docs/docs/index.rst)\n\nSearch \n     \n\n* * *\n\nHosted by
     [ Read the Docs ](https://readthedocs.org) \u00b7 [ Privacy Policy ](https://
     docs.readthedocs.io/page/privacy-policy.html)\n",
46     "extras": {
47       "url": "https://neurokit2.readthedocs.io/en/legacy_docs/",
48       ...... (webpage information omitted),
49     }
50   }
51 ],
52 [
53   {
54     "id": 22,
55     "timestamp": "2024-09-19T14:13:58.344923",
56     "source": "agent",
57     "message": "Executing browser actions: click('43')\nfill('43', 'rsp_rrv')\
     npress('43', 'Enter')",
58     "action": "browse_interactive",
59     "args": {
60       "browser_actions": "click('43')\nfill('43', 'rsp_rrv')\npress('43', 'Enter')
     ",
61       "thought": "To find the correct usage of the 'rsp_rrv' function, I should
     search for it in the NeuroKit2 documentation. The current page has a search box
     that I can use for this purpose.\n\nAction:\n1. Click on the search textbox.\n2.
      Fill the search textbox with the query \"rsp_rrv\".\n3. Press Enter to initiate
      the search.",
62       "browsergym_send_msg_to_user": ""
63     }
64   },
65   {
66     "id": 23,
67     "timestamp": "2024-09-19T14:14:02.341831",
68     "source": "agent",
69     "message": "Visited https://neurokit2.readthedocs.io/en/legacy_docs/search.html
     ?q=rsp_rrv&check_keywords=yes&area=default",
70     "cause": 22,
71     "observation": "browse",
72     "content": "[ <::before> NeuroKit  ](index.html)\n\nlegacy_docs \n\n__ [
     NeuroKit ](index.html)\n\n<::before><::after> \n  * [ Docs ](index.html) \u00bb \
     n  * Search \n  * [ <::before> Edit on GitHub ](https://github.com/
     neuropsychology/NeuroKit/blob/legacy_docs/docs/search.rst)\n\n\n\n* * *\n\
     nPlease activate JavaScript to enable the search functionality. \n\n## 
     Searching   .. \n\n\n\n\n* * *\n\n\u00a9 Copyright 2020, Dominique Makowski
     Revision ' bd105f7f ' . \n\n<::before><::after> <::before> Read the Docs  v:
     legacy_docs  <::before>\n\nVersions \n     [ legacy_docs ](https://neurokit2.
     readthedocs.io/en/legacy_docs/search.html)\n\nDownloads \n     [ PDF ](//
     neurokit2.readthedocs.io/_/downloads/en/legacy_docs/pdf/)\n     [ HTML ](//
     neurokit2.readthedocs.io/_/downloads/en/legacy_docs/htmlzip/)\n     [ Epub ](//
     neurokit2.readthedocs.io/_/downloads/en/legacy_docs/epub/)\n\nOn Read the Docs \
     n     [ Project Home ](//readthedocs.org/projects/neurokit2/)\n     [ Builds
     ](//readthedocs.org/projects/neurokit2/builds/)\n     [ Downloads ](//
     readthedocs.org/projects/neurokit2/downloads/)\n\nOn GitHub \n     [ View ](
     https://github.com/neuropsychology/NeuroKit/blob/legacy_docs/docs/search.rst)\n
        [ Edit ](https://github.com/neuropsychology/NeuroKit/edit/legacy_docs/docs/
     search.rst)\n\nSearch \n     \n\n* * *\n\nHosted by [ Read the Docs ](https://
     readthedocs.org) \u00b7 [ Privacy Policy ](https://docs.readthedocs.io/page/
     privacy-policy.html)\n",
73       "extras": {
74         "url": "https://neurokit2.readthedocs.io/en/legacy_docs/search.html?q=rsp_rrv
     &check_keywords=yes&area=default",
75         ...... (webpage information omitted),
76       }
77     }
78 ],
79 [
80   {
81     "id": 24,
82     "timestamp": "2024-09-19T14:14:03.570167",
83     "source": "agent",
84     "message": "Executing browser actions: scroll(0, 300)",
85     "action": "browse_interactive",
86     "args": {
```

```
87          "browser_actions": "scroll(0, 300)",
88          "thought": "To find the correct usage of the `rsp_rrv` function, I need to
         review the search results. Since the search results are likely listed in the
         main section, I will scroll down to view them.",
89          "browsergym_send_msg_to_user": ""
90        }
91      },
92      {
93        "id": 25,
94        "timestamp": "2024-09-19T14:14:05.356163",
95        "source": "agent",
96        "message": "Visited https://neurokit2.readthedocs.io/en/legacy_docs/search.html
         ?q=rsp_rrv&check_keywords=yes&area=default",
97        "cause": 24,
98        "observation": "browse",
99        "content": "[ <::before> NeuroKit  ](index.html)\n\nlegacy_docs \n\n__ [
         NeuroKit ](index.html)\n\n<::before><::after>\n  * [ Docs ](index.html) \u00bb \
         n  * Search \n  * [ <::before> Edit on GitHub ](https://github.com/
         neuropsychology/NeuroKit/blob/legacy_docs/docs/search.rst)\n\n\n\n* *\n\
         nPlease activate JavaScript to enable the search functionality. \n\n##
         Searching \n\n\n\n\n* * *\n\n\u00a9 Copyright 2020, Dominique Makowski  Revision
          ` bd105f7f ` . \n\n<::before><::after> <::before> Read the Docs  v: legacy_docs
          <::before>\n\nVersions \n    [ legacy_docs ](https://neurokit2.readthedocs.io
         /en/legacy_docs/search.html)\n\nDownloads \n    [ PDF ](//neurokit2.readthedocs
         .io/_/downloads/en/legacy_docs/pdf/)\n    [ HTML ](//neurokit2.readthedocs.io/_
         /downloads/en/legacy_docs/htmlzip/)\n    [ Epub ](//neurokit2.readthedocs.io/_/
         downloads/en/legacy_docs/epub/)\n\nOn Read the Docs \n    [ Project Home ](//
         readthedocs.org/projects/neurokit2/)\n    [ Builds ](//readthedocs.org/projects
         /neurokit2/builds/)\n    [ Downloads ](//readthedocs.org/projects/neurokit2/
         downloads/)\n\nOn GitHub \n    [ View ](https://github.com/neuropsychology/
         NeuroKit/blob/legacy_docs/docs/search.rst)\n    [ Edit ](https://github.com/
         neuropsychology/NeuroKit/edit/legacy_docs/docs/search.rst)\n\nSearch \n    \n\n*
          * *\n\nHosted by [ Read the Docs ](https://readthedocs.org) \u00b7 [ Privacy
         Policy ](https://docs.readthedocs.io/page/privacy-policy.html)\n",
100       "extras": {
101         "url": "https://neurokit2.readthedocs.io/en/legacy_docs/search.html?q=rsp_rrv
         &check_keywords=yes&area=default",
102         ...... (webpage information omitted),
103       }
104     }
105   ],
106   ......(19 more turns omitted),
107 }
```

Listing F.2: An example of Claude-3.5-Sonnet that struggles with specialized bash commands in
OpenHands to edit programs correctly. See line 11 for a failed edit example.

```
1 {
2   ......(first seven turns omitted),
3   [
4     {
5       "id": 15,
6       "timestamp": "2024-09-16T21:45:01.776300",
7       "source": "agent",
8       "message": "......",
9       "action": "run_ipython",
10      "args": {
11        "code": "code = '''\nimport pandas as pd\nimport numpy as np\nfrom sklearn.
         model_selection import train_test_split\nfrom sklearn.preprocessing import
         StandardScaler, OneHotEncoder\nfrom sklearn.linear_model import
         LogisticRegression\nfrom sklearn.metrics import accuracy_score\nfrom sklearn.
         compose import ColumnTransformer\nfrom sklearn.pipeline import Pipeline\nfrom
         mlxtend.feature_selection import SequentialFeatureSelector as SFS\nimport
         matplotlib.pyplot as plt\n\n# Load the data\ndata = pd.read_csv('/workspace/
         benchmark/datasets/dkpes/dkpes_train.csv')\n\n# Prepare features and target\nX =
          data.drop(['index', 'Signal-inhibition'], axis=1)\ny = data['Signal-inhibition
         ']\n\n# Identify numeric and categorical columns\nnumeric_features = X.
         select_dtypes(include=['int64', 'float64']).columns\ncategorical_features = X.
         select_dtypes(include=['object']).columns\n\n# Binarize the target variable\
         nthreshold = y.median()\ny_binary = (y > threshold).astype(int)\n\n# Split the
         data\nX_train, X_test, y_train, y_test = train_test_split(X, y_binary, test_size
         =0.2, random_state=42)\n\n# Create preprocessing steps\npreprocessor =
         ColumnTransformer(\n    transformers=[\n        ('num', StandardScaler(),
         numeric_features),\n        ('cat', OneHotEncoder(drop='first', sparse=False),
         categorical_features)\n    ])\n\n# Create a pipeline with preprocessing and
         logistic regression\nlr_pipeline = Pipeline([\n    ('preprocessor', preprocessor
         ),\n    ('classifier', LogisticRegression(random_state=42))\n])\n\n# Perform
         backward feature selection\nsfs = SFS(lr_pipeline,\n        k_features=(1, len
         (X.columns)),\n        forward=False,\n        floating=False,\n
         scoring='accuracy',\n        cv=5,\n        n_jobs=-1)\n\nsfs = sfs.fit(
```

```
        X_train, y_train)\n\n# Get the results\nresults = pd.DataFrame.from_dict(sfs.
        get_metric_dict()).T\nresults.sort_index(inplace=True)\n\n# Plot the results\
        nplt.figure(figsize=(10, 6))\nplt.plot(results.index, results['avg_score'],
        marker='o')\nplt.title('Backward Feature Selection: Accuracy vs Number of
        Features')\nplt.xlabel('Number of Features')\nplt.ylabel('Cross-validation
        Accuracy')\nplt.grid(True)\nplt.tight_layout()\n\n# Save the plot\nplt.savefig
        ('/workspace/pred_results/dkpes_feature_selection_analysis_pred.png')\n\n# Print
        the best feature subset\nbest_subset = list(sfs.k_feature_names_)\nprint(f\"
        Best feature subset: {best_subset}\")\nprint(f\"Number of selected features: {
        len(best_subset)}\")\nprint(f\"Best cross-validation accuracy: {sfs.k_score_:.4f
        }\")\n\n# Evaluate on the test set using the best feature subset\nbest_pipeline
        = sfs.estimator\nbest_pipeline.fit(X_train, y_train)\ny_pred = best_pipeline.
        predict(X_test)\ntest_accuracy = accuracy_score(y_test, y_pred)\nprint(f\"Test
        accuracy with selected features: {test_accuracy:.4f}\")\n'''\n\
        nedit_file_by_replace('/workspace/pred_programs/pred_dkpes_visualization_3.py',
        \n                    to_replace='import pandas as pd\\nimport numpy as np\\
        nfrom sklearn.model_selection import train_test_split\\nfrom sklearn.
        preprocessing import StandardScaler, OneHotEncoder\\nfrom sklearn.linear_model
        import LogisticRegression\\nfrom sklearn.metrics import accuracy_score\\nfrom
        sklearn.compose import ColumnTransformer\\nfrom sklearn.pipeline import Pipeline
        \\nfrom mlxtend.feature_selection import SequentialFeatureSelector as SFS\\
        nimport matplotlib.pyplot as plt', \n                          new_content=code)",
          "thought": "Thank you for updating the file. I see that there are some
        duplicate sections in the code. Let me clean it up and remove the unnecessary
        parts:",
          "is_confirmed": "confirmed",
          "kernel_init_code": "from agentskills import *"
        }
      },
      {
        "id": 16,
        "timestamp": "2024-09-16T21:45:01.983049",
        "source": "agent",
        "message": "Code executed in IPython cell.",
        "cause": 15,
        "observation": "run_ipython",
        "content": "......",
        "extras": {
          "code": "......"
        }
      }
    ],
    [
      {
        "id": 17,
        "timestamp": "2024-09-16T21:45:20.373176",
        "source": "agent",
        "message": "......",
        "action": "run_ipython",
        "args": {
          "code": "......",
          "thought": "I apologize for the confusion. It seems that the file still
        contains duplicate content. Let me clean it up one more time to remove all
        duplicates and ensure the code is correct:",
          "is_confirmed": "confirmed",
          "kernel_init_code": "from agentskills import *"
        }
      },
      {
        "id": 18,
        "timestamp": "2024-09-16T21:45:20.397554",
        "source": "agent",
        "message": "Code executed in IPython cell.",
        "cause": 17,
        "observation": "run_ipython",
        "content": "  Cell In[1], line 88\n    new_content=\n                     ^\
        nSyntaxError: incomplete input\n[Jupyter current working directory: /workspace/
        benchmark/datasets]\n[Jupyter Python interpreter: /openhands/poetry/openhands-ai
        -5O4_aCHf-py3.11/bin/python]",
        "extras": {
          "code": "......"
        }
      }
    ],
    [
      {
        "id": 19,
        "timestamp": "2024-09-16T21:45:35.261112",
        "source": "agent",
        "message": "......",
        "action": "run_ipython",
```

```
 64        "args": {
 65          "code": "......",
 66          "thought": "I apologize for the syntax error. Let me correct it and update
        the file properly:",
 67          "is_confirmed": "confirmed",
 68          "kernel_init_code": "from agentskills import *"
 69        }
 70      },
 71      {
 72        "id": 20,
 73        "timestamp": "2024-09-16T21:45:35.480240",
 74        "source": "agent",
 75        "message": "Code executed in IPython cell.",
 76        "cause": 19,
 77        "observation": "run_ipython",
 78        "content": "......",
 79        "extras": {
 80          "code": "......"
 81        }
 82      }
 83    ],
 84    [
 85      {
 86        "id": 21,
 87        "timestamp": "2024-09-16T21:45:52.166468",
 88        "source": "agent",
 89        "message": "......",
 90        "action": "run_ipython",
 91        "args": {
 92          "code": "......",
 93          "thought": "I apologize for the confusion. It seems that the file still
        contains duplicate content. Let me clean it up one final time to remove all
        duplicates and ensure the code is correct:",
 94          "is_confirmed": "confirmed",
 95          "kernel_init_code": "from agentskills import *"
 96        }
 97      },
 98      {
 99        "id": 22,
100        "timestamp": "2024-09-16T21:45:52.190187",
101        "source": "agent",
102        "message": "Code executed in IPython cell.",
103        "cause": 21,
104        "observation": "run_ipython",
105        "content": "  Cell In[1], line 88\n    new_content=\n                       ^\
        nSyntaxError: incomplete input\n[Jupyter current working directory: /workspace/
        benchmark/datasets]\n[Jupyter Python interpreter: /openhands/poetry/openhands-ai
        -5O4_aCHf-py3.11/bin/python]",
106        "extras": {
107          "code": "......"
108        }
109      }
110    ],
111    [
112      {
113        "id": 23,
114        "timestamp": "2024-09-16T21:46:10.802743",
115        "source": "agent",
116        "message": "......",
117        "action": "run_ipython",
118        "args": {
119          "code": "......",
120          "thought": "I apologize for the repeated mistake. Let me correct it and
        update the file properly:",
121          "is_confirmed": "confirmed",
122          "kernel_init_code": "from agentskills import *"
123        }
124      },
125      {
126        "id": 24,
127        "timestamp": "2024-09-16T21:46:11.030574",
128        "source": "agent",
129        "message": "Code executed in IPython cell.",
130        "cause": 23,
131        "observation": "run_ipython",
132        "content": "......",
133        "extras": {
134          "code": "......"
135        }
136      }
137    ],
```

```
138  [
139    {
140      "id": 25,
141      "timestamp": "2024-09-16T21:46:26.874027",
142      "source": "agent",
143      "message": "......",
144      "action": "run_ipython",
145      "args": {
146        "code": "with open('/workspace/pred_programs/pred_dkpes_visualization_3.py',
         'w') as f:\n    f.write('''\nimport pandas as pd\nimport numpy as np\nfrom
         sklearn.model_selection import train_test_split\nfrom sklearn.preprocessing
         import StandardScaler, OneHotEncoder\nfrom sklearn.linear_model import
         LogisticRegression\nfrom sklearn.metrics import accuracy_score\nfrom sklearn.
         compose import ColumnTransformer\nfrom sklearn.pipeline import Pipeline\nfrom
         mlxtend.feature_selection import SequentialFeatureSelector as SFS\nimport
         matplotlib.pyplot as plt\n\n# Load the data\ndata = pd.read_csv('/workspace/
         benchmark/datasets/dkpes/dkpes_train.csv')\n\n# Prepare features and target\nX =
          data.drop(['index', 'Signal-inhibition'], axis=1)\ny = data['Signal-inhibition
         ']\n\n# Identify numeric and categorical columns\nnumeric_features = X.
         select_dtypes(include=['int64', 'float64']).columns\ncategorical_features = X.
         select_dtypes(include=['object']).columns\n\n# Binarize the target variable\
         nthreshold = y.median()\ny_binary = (y > threshold).astype(int)\n\n# Split the
         data\nX_train, X_test, y_train, y_test = train_test_split(X, y_binary, test_size
         =0.2, random_state=42)\n\n# Create preprocessing steps\npreprocessor =
         ColumnTransformer(\n    transformers=[\n        ('num', StandardScaler(),
         numeric_features),\n        ('cat', OneHotEncoder(drop='first', sparse=False),
         categorical_features)\n    ])\n\n# Create a pipeline with preprocessing and
         logistic regression\nlr_pipeline = Pipeline([\n    ('preprocessor', preprocessor
         ),\n    ('classifier', LogisticRegression(random_state=42))])\n\n# Perform
         backward feature selection\nsfs = SFS(lr_pipeline,\n        k_features=(1, len
         (X.columns)),\n        forward=False,\n        floating=False,\n
         scoring='accuracy',\n        cv=5,\n        n_jobs=-1)\n\nsfs = sfs.fit(
         X_train, y_train)\n\n# Get the results\nresults = pd.DataFrame.from_dict(sfs.
         get_metric_dict()).T\nresults.sort_index(inplace=True)\n\n# Plot the results\
         nplt.figure(figsize=(10, 6))\nplt.plot(results.index, results['avg_score'],
         marker='o')\nplt.title('Backward Feature Selection: Accuracy vs Number of
         Features')\nplt.xlabel('Number of Features')\nplt.ylabel('Cross-validation
         Accuracy')\nplt.grid(True)\nplt.tight_layout()\n\n# Save the plot\nplt.savefig
         ('/workspace/pred_results/dkpes_feature_selection_analysis_pred.png')\n\n# Print
          the best feature subset\nbest_subset = list(sfs.k_feature_names_)\nprint(f\"
         Best feature subset: {best_subset}\")\nprint(f\"Number of selected features: {
         len(best_subset)}\")\nprint(f\"Best cross-validation accuracy: {sfs.k_score_:.4f
         }\")\n\n# Evaluate on the test set using the best feature subset\nbest_pipeline
         = sfs.estimator\nbest_pipeline.fit(X_train, y_train)\ny_pred = best_pipeline.
         predict(X_test)\ntest_accuracy = accuracy_score(y_test, y_pred)\nprint(f\"Test
          accuracy with selected features: {test_accuracy:.4f}\")\n''')\n\nprint(\"File
         content updated successfully.\")",
147        "thought": "I apologize for the persistent issue. Let me fix this once and
         for all by directly editing the file content:",
148        "is_confirmed": "confirmed",
149        "kernel_init_code": "from agentskills import *"
150      }
151    },
152    {
153      "id": 26,
154      "timestamp": "2024-09-16T21:46:26.901372",
155      "source": "agent",
156      "message": "Code executed in IPython cell.",
157      "cause": 25,
158      "observation": "run_ipython",
159      "content": "File content updated successfully.\n[Jupyter current working
         directory: /workspace/benchmark/datasets]\n[Jupyter Python interpreter: /
         openhands/poetry/openhands-ai-5O4_aCHf-py3.11/bin/python]",
160      "extras": {
161        "code": "......"
162      }
163    }
164  ],
165  ......(17 more turns omitted)
166  }
```

## F.2 CASE 2: INFLUENCE OF EXPERT-PROVIDED KNOWLEDGE

In Section 4, we also find that expert-provided knowledge does not always lead to metric improvement because: (1) Expert-provided knowledge specifies some advanced tools that are less familiar to the agents. (2) When the agents do not know how to solve some tasks without domain knowledge, they tend to generate some executable but less meaningful programs.

As show in Listing F.3, without expert-provided knowledge, Claude-3.5-Sonnet using self-debug would only use basic tools like pandas and sklearn to develop a random forest model, which is not accurate enough to find the best drugs to repurpose. To help the agent, an expert provided the following knowledge: "Typical encoders for a drug include ECFP fingerprint or message-passing neural networks over the 2D molecular graph representation and for a target include 1D convolutional neural networks over the amino acid sequence. " With the help of expert-provided knowledge, the same agent successfully generates a program to install the DeepPurpose package and use it do develop an advanced neural network with MPNN drug encoding and CNN target encoding (lines47–57 in Listing F.4). However, we noticed some signs of data contamination in this generated program, such as the use of automatic data loader in DeepPurpose (lines 33-38 in Listing F.4). This issue is effectively captured by our strategies to mitigate data contamination and agent shortcut concerns (Section 2.2): The generated program (Listing F.4) is not executable as the automatic data loader does not read the modified local dataset correctly. Despite this issue, from a scientist user's perspective, the program generated with expert-provided knowledge tries to develop a more helpful model and resembles the ground-truth program (Listing F.5) more closely.

Listing F.3: Program generated by Claude-3.5-Sonnet using self-debug **without** expert-provided knowledge.

```python
1  import numpy as np
2  import pandas as pd
3  from sklearn.model_selection import train_test_split
4  from sklearn.ensemble import RandomForestRegressor
5  from sklearn.preprocessing import StandardScaler
6  import json
7  import os
8
9  # Load and preprocess data
10 def load_data(file_path):
11     with open(file_path, 'r') as f:
12         return [line.strip() for line in f]
13
14 def load_affinity_data(file_path):
15     return pd.read_csv(file_path, header=None).values
16
17 def load_target_sequences(file_path):
18     with open(file_path, 'r') as f:
19         return json.load(f)
20
21 def encode_sequence(seq, max_length=100):
22     amino_acids = 'ACDEFGHIKLMNPQRSTVWY'
23     encoding = np.zeros(max_length)
24     for i, aa in enumerate(seq[:max_length]):
25         if aa in amino_acids:
26             encoding[i] = amino_acids.index(aa) + 1
27     return encoding
28
29 # Load data
30 drugs_train = load_data('benchmark/datasets/dti/DAVIS/drug_train.txt')
31 affinity_train = load_affinity_data('benchmark/datasets/dti/DAVIS/affinity_train.csv'
       )
32 target_sequences = load_target_sequences('benchmark/datasets/dti/DAVIS/target_seq.
       json')
33
34 # Encode drugs (simple fingerprint)
35 def encode_drug(smiles):
36     return [smiles.count(c) for c in 'CNOFPS']
37
38 X_drugs = np.array([encode_drug(drug) for drug in drugs_train])
39 X_targets = np.array([encode_sequence(seq) for seq in target_sequences.values()])
40
41 # Prepare data for training
42 X = np.column_stack([np.repeat(X_drugs, X_targets.shape[0], axis=0),
43                      np.tile(X_targets, (X_drugs.shape[0], 1))])
44 y = affinity_train.flatten()
```

```
45
46  # Scale features
47  scaler = StandardScaler()
48  X_scaled = scaler.fit_transform(X)
49
50  # Split data
51  X_train, X_test, y_train, y_test = train_test_split(X_scaled, y, test_size=0.2,
        random_state=42)
52
53  # Train model
54  model = RandomForestRegressor(n_estimators=50, max_depth=10, n_jobs=-1, random_state
        =42)
55  model.fit(X_train, y_train)
56
57  # Load COVID-19 target and antiviral drugs
58  covid_seq = load_data('benchmark/datasets/dti/covid_seq.txt')[1]  # Second line
        contains the sequence
59  antiviral_drugs = pd.read_csv('benchmark/datasets/dti/antiviral_drugs.tab', sep='\t')
60
61  # Predict binding affinities
62  covid_encoding = encode_sequence(covid_seq)
63  X_pred = np.column_stack([np.array([encode_drug(smiles) for smiles in antiviral_drugs
        ['SMILES']]),
64                           np.tile(covid_encoding, (len(antiviral_drugs), 1))])
65  X_pred_scaled = scaler.transform(X_pred)
66
67  predictions = model.predict(X_pred_scaled)
68
69  # Rank drugs based on predicted affinities
70  ranked_drugs = antiviral_drugs.copy()
71  ranked_drugs['Predicted_Affinity'] = predictions
72  ranked_drugs = ranked_drugs.sort_values('Predicted_Affinity')
73
74  # Save results
75  os.makedirs('pred_results', exist_ok=True)
76  with open('pred_results/davis_dti_repurposing.txt', 'w') as f:
77      for smiles in ranked_drugs['SMILES']:
78          f.write(f"{smiles}\n")
79
80  print("Predictions saved to pred_results/davis_dti_repurposing.txt")
```

Listing F.4: Program generated by Claude-3.5-Sonnet using self-debug **with** expert-provided knowledge.

```
1  import os
2  import sys
3  import subprocess
4
5  # Function to install required packages using pip
6  def install_pip(package):
7      subprocess.check_call([sys.executable, "-m", "pip", "install", package])
8
9  # Check and install required packages
10 required_packages = ['pandas', 'rdkit', 'DeepPurpose', 'git+https://github.com/bp-
        kelley/descriptastorus', 'pandas-flavor']
11 for package in required_packages:
12     try:
13         if package.startswith('git+'):
14             install_pip(package)
15         else:
16             __import__(package.split('==')[0])
17     except ImportError:
18         print(f"{package} not found. Installing...")
19         if package == 'rdkit':
20             install_pip('rdkit-pypi')  # Use rdkit-pypi for pip installation
21         else:
22             install_pip(package)
23
24 # Now that we've ensured all packages are installed, import the required modules
25 import numpy as np
26 import pandas as pd
27 from DeepPurpose import DTI as models
28 from DeepPurpose import utils, dataset
29
30 # Create directories if they don't exist
31 os.makedirs('pred_results', exist_ok=True)
32
33 # Load and preprocess DAVIS dataset
34 X_drug, X_target, y = dataset.load_process_DAVIS(
35     path='benchmark/datasets/dti/DAVIS',
```

```
36      binary=False,
37      convert_to_log=True
38  )
39
40  # Convert numpy arrays to pandas DataFrames
41  train = pd.DataFrame({
42      'drug': X_drug,
43      'target': X_target,
44      'Label': y  # Changed 'Y' to 'Label'
45  })
46
47  # Load drug-target interaction model
48  config = utils.generate_config(
49      drug_encoding='MPNN',
50      target_encoding='CNN',
51      cls_hidden_dims=[1024, 1024, 512],
52      train_epoch=5
53  )
54  model = models.model_initialize(**config)
55
56  # Train the model
57  model.train(train)
58
59  # Load antiviral drugs
60  antiviral_drugs = pd.read_csv('benchmark/datasets/dti/antiviral_drugs.tab', sep='\t')
61
62  # Load COVID-19 target sequence
63  with open('benchmark/datasets/dti/covid_seq.txt', 'r') as f:
64      covid_seq = f.read().strip().split('\n')[0]
65
66  # Prepare data for prediction
67  X_drug_repurpose = antiviral_drugs['SMILES'].tolist()
68  X_target_repurpose = [covid_seq] * len(X_drug_repurpose)
69
70  # Make predictions
71  y_pred = model.predict(X_drug_repurpose, X_target_repurpose)
72
73  # Create a dataframe with predictions
74  results = pd.DataFrame({
75      'Drug': antiviral_drugs['Name'],
76      'SMILES': antiviral_drugs['SMILES'],
77      'Predicted_Affinity': y_pred
78  })
79
80  # Sort by predicted affinity (lower values indicate stronger binding)
81  results_sorted = results.sort_values('Predicted_Affinity')
82
83  # Save results
84  results_sorted['SMILES'].to_csv('pred_results/davis_dti_repurposing.txt', index=False
        , header=False)
85
86  print("Prediction completed. Results saved to pred_results/davis_dti_repurposing.txt"
        )
```

Listing F.5: Ground-truth program in the benchmark.

```
1  from DeepPurpose import utils, dataset
2  from DeepPurpose import DTI as models
3  from pathlib import Path
4  from shutil import copyfile
5
6  import os
7  import json
8  import numpy as np
9  import pandas as pd
10
11  drug_encoding, target_encoding = 'MPNN', 'CNN'
12
13  def make_dataset(drug_fname, affinity_fname, target):
14      with open(drug_fname) as f:
15          drug = [l.rstrip() for l in f]
16
17      affinity = pd.read_csv(affinity_fname, header=None)
18
19      SMILES = []
20      Target_seq = []
21      y = []
22
23      for i in range(len(drug)):
24          for j in range(len(target)):
```

```
25            SMILES.append(drug[i])
26            Target_seq.append(target[j])
27            y.append(affinity.values[i, j])
28
29     y = utils.convert_y_unit(np.array(y), 'nM', 'p')
30
31     return utils.data_process(np.array(SMILES), np.array(Target_seq), np.array(y),
32                               drug_encoding, target_encoding,
33                               split_method='no_split')
34
35
36
37 def main():
38     with open('benchmark/datasets/dti/DAVIS/target_seq.json') as f:
39         target = json.load(f)
40     target = list(target.values())
41
42     train = make_dataset('benchmark/datasets/dti/DAVIS/drug_train.txt', 'benchmark/
        datasets/dti/DAVIS/affinity_train.csv', target)
43     val = make_dataset('benchmark/datasets/dti/DAVIS/drug_val.txt', 'benchmark/
        datasets/dti/DAVIS/affinity_val.csv', target)
44
45     config = utils.generate_config(drug_encoding = drug_encoding,
46                           target_encoding = target_encoding,
47                           cls_hidden_dims = [1024,1024,512],
48                           train_epoch = 10,
49                           LR = 5e-4,
50                           batch_size = 128,
51                           hidden_dim_drug = 128,
52                           mpnn_hidden_size = 128,
53                           mpnn_depth = 3,
54                           cnn_target_filters = [32,64,96],
55                           cnn_target_kernels = [4,8,12]
56                           )
57
58     model = models.model_initialize(**config)
59
60     model.train(train, val, val)
61
62     t, t_name = [l.rstrip() for l in open('benchmark/datasets/dti/covid_seq.txt')]
63
64     df = pd.read_csv('benchmark/datasets/dti/antiviral_drugs.tab', sep = '\t')
65     r, r_name, r_pubchem_cid = df.SMILES.values, df['Name'].values, df['Pubchem CID'
        ].values
66
67     out_fpath = Path("./pred_results/result/")
68     if not out_fpath.exists():
69         os.mkdir(out_fpath)
70
71     y_pred = models.repurpose(X_repurpose = r, target = t, model = model, drug_names
         = r_name, target_name = t_name,
72                               result_folder = "./pred_results/result/", convert_y =
         True)
73
74     with open("./pred_results/result/repurposing.txt") as f_in:
75         lines = [l for l in f_in]
76
77     with open("./pred_results/davis_dti_repurposing.txt", "w+") as f_out:
78         f_out.write("".join(lines[3:-1]))
79
80
81 if __name__ == "__main__":
82     main()
```

# G EXPERT VALIDATION DETAILS

In this section, we provide details about the expert validation process in Section 2.2. We include the questionnaire (Section G.1) for domain experts and two examples used in it (Section G.2 and G.3).

## G.1 QUESTIONNAIRE FOR DOMAIN EXPERTS

Thanks for providing feedback on our AI4Science benchmark called ScienceAgentBench. We are developing an AI agent to assist you! Given a task instruction and a dataset, the agent will help you write a computer program to fulfill the task you have in mind. To develop and evaluate such an AI agent, we have collected a benchmark by adapting some tasks from peer-reviewed publications with open-source codes. Each data sample in our benchmark consists of the following main components:

**Task Instruction**: Describes (1) the goal of a task or a scientific hypothesis and (2) output requirements.

**Dataset Information**: Contains (1) the dataset directory structure and (2) helpful metadata or a few examples from the dataset.

**Annotated Program**: The reference solution adapted from each publication's open-source code.

**Evaluation Script**: The code to evaluate AI agents' performance by comparing the execution results of its generated programs with those of the annotated programs.

To ensure that each task is formulated and described correctly and professionally, we would like you to give us a hand by reviewing our collected data samples. In addition, we are also seeking some additional information from you as a domain expert, including writing down some task-related domain knowledge and revising a rubric to score the generated programs.

Please follow the guidelines below to review each task. First, please enter the Task ID you are reviewing: [task_id]

**Guidelines for Data Reviewing**

First, a bit more background: once you give the AI agent a task instruction, it will try to automatically complete everything without seeking additional help from you. This is similar to the scenario where you give the task to a junior student in your lab/class who will complete it as an assignment.

For each task, please first spend a few minutes reading the given task information (instruction, dataset information, and source GitHub repository) and our annotated program to have a rough understanding of the task and relevant concepts. Then, please comment on the following two parts. Note that you may iteratively revise your answer to each question to help us improve the task instructions and programs.

**1. Program**

Is the program a valid solution (not necessarily the best solution) to the given task instruction? Here is an example: [google_doc_link] [a]

If there are only minor issues, please comment on how the program should be modified below. However, if you believe there is a major issue (e.g., the program is doing sth irrelevant or more than two lines of code need to be revised in order to make it correct), please let us know the task ID and do NOT fill the rest of the form.

Is the program a valid solution to the given task instruction?

[] Yes

[] Need Modification (comment below)

[] No (report and continue to the next task)

How should the program be modified? Please mention the line numbers that need to be inspected.

[Long Text Answer]

**2. Task Instruction**

The task instructions were created by non-experts and thus might contain some misused terms, awkward expressions, or inaccurate descriptions of the task that do not adhere to your domain's scientific language. Do you see such issues for this task instruction? If so, please revise or rewrite the task instruction for any issues you can find.

Finally, if needed, please help make the task instruction more fluent and natural sounding.

Please enter your revised task instruction below. If there are no changes, please skip this question and leave the answer text blank.

[Long Text Answer]

---

[a]The example for program and instruction validation is provided in Section G.2.

**3. Domain Knowledge**

Suppose the AI agent fails to fulfill the task based solely on the task instruction, perhaps due to lack of some background knowledge, we want to provide some additional information to help it succeed. This is similar to the situation where you give an exam problem to a student in your class and they might not be able to do it just based on the problem description, but you can provide some hints to help them. Please write down at most three important pieces of knowledge that are related to the task and the program.

For example ([google_doc_link]): [a]

Concepts and details in the task description or program that may need further explanation or extra attention, e.g., a term definition on wikipedia you would send to a new student (without much domain expertise) working on this task, or a common practice for such tasks in your field.

Information about the python packages and/or functions used in the program. For example, you would copy and paste a snippet of package description/function documentation to help the new student working on the task.

You may assume the AI agent has a general sense of your domain, like a new graduate student with undergrad-level knowledge but not much about the specific task. Please help it by providing some knowledge to write the program for this task. You can search online for more details about the dataset, packages, and functions used in each task before writing.

Please try not to "leak" the annotated program directly. You may imagine that you don't have the direct answer but could provide some helpful information to your junior colleague so that they can derive the program. For example:

Instead of copying/describing a few lines in the program, you may copy the documentations of packages/functions used in that program.

Instead of specifying the variables and parameters, you may suggest a range (e.g., 1e-3 to 1e-4 for learning rate).

Instead of saying columns A,B,C are related to the target attribute Y, you may try to find a knowledge snippet describing what is correlated to Y.

However, in some rare cases, there may be a need to provide a minimal "leak" of the annotated program, e.g., the decision boundary of Y is 0.6 instead of 0.5. Still, it would be great if you could think about its necessity before annotating such knowledge.

For each piece of domain knowledge related to this task, please write 1-5 sentences. If you believe the task instruction is self-contained and needs no further explanations, please enter "None".

[Long Text Answer]

**4. Scoring Rubric**

Once the AI agent generates a program, we need an evaluation method to review the generated program. To do it, we need a task-specific scoring rubric, which assigns partial credits for more comprehensive evaluation of the generated program. Right now we have already got an initial draft of the rubric with five major components: (1) data loading, (2) data processing, (3) modeling, analysis or visualization, (4) output formatting, (5) saving output.

Please review our initial draft of the rubric. Imagine that you will use this rubric to score the programs produced by your junior students. Please modify the rubric items that you think are incorrect, should be described with more/less details, or should be reweighed with higher/lower credits for each component. Please also add any missing but necessary rubric item you would use to assess a program's correctness, or remove redundant rubric items.

Please enter your revised scoring rubric below. If there are no changes, please skip this question and leave the answer text blank.

[Long Text Answer]

---

[a]The example for domain knowledge annotation is provided in Section G.3.

## G.2 PROGRAM EXAMPLE FOR DOMAIN EXPERTS

**Task Instruction:** Train a graph convolutional network on the given dataset to predict the aquatic toxicity of compounds. Use the resulting model to compute and visualize the atomic contributions to molecular activity of the given test example compound. Save the figure as "pred_results/aquatic_toxicity_qsar_vis.png".

**Program:**

```python
import os
os.environ["TF_USE_LEGACY_KERAS"] = "1"

from rdkit import Chem
from rdkit.Chem.Draw import SimilarityMaps

import pandas as pd
import deepchem as dc

def vis_contribs(mol, df, smi_or_sdf = "sdf"):
    wt = {}
    if smi_or_sdf == "smi":
        for n,atom in enumerate(
            Chem.rdmolfiles.CanonicalRankAtoms(mol)
        ):
            wt[atom] = df.loc[mol.GetProp("_Name"),"Contrib"][n]
    if smi_or_sdf == "sdf":
        for n,atom in enumerate(range(mol.GetNumHeavyAtoms())):
            wt[atom] = df.loc[Chem.MolToSmiles(mol),"Contrib"][n]
    return SimilarityMaps.GetSimilarityMapFromWeights(mol,wt)

def main():
    DATASET_FILE = os.path.join(
        'benchmark/datasets/aquatic_toxicity',
        'Tetrahymena_pyriformis_OCHEM.sdf'
    )

    mols = [
        m
        for m in Chem.SDMolSupplier(DATASET_FILE)
        if m is not None
    ]
    loader = dc.data.SDFLoader(
        tasks=["IGC50"],
        featurizer=dc.feat.ConvMolFeaturizer(),
        sanitize=True
    )
    dataset = loader.create_dataset(DATASET_FILE, shard_size=5000)

    m = dc.models.GraphConvModel(
        1,
        mode="regression",
        batch_normalize=False
    )
    m.fit(dataset, nb_epoch=40)

    TEST_DATASET_FILE = os.path.join(
        'benchmark/datasets/aquatic_toxicity',
        'Tetrahymena_pyriformis_OCHEM_test_ex.sdf'
    )
    test_mol = [
        m
        for m in Chem.SDMolSupplier(TEST_DATASET_FILE)
        if m is not None
    ][0]
    test_dataset = loader.create_dataset(
        TEST_DATASET_FILE,
        shard_size=5000
    )

    loader = dc.data.SDFLoader(
        tasks=[],
        featurizer=dc.feat.ConvMolFeaturizer(
            per_atom_fragmentation=True
        ),
        sanitize=True
    )
    frag_dataset = loader.create_dataset(
        TEST_DATASET_FILE,
        shard_size=5000
```

```
71          )
72
73          tr = dc.trans.FlatteningTransformer(frag_dataset)
74          frag_dataset = tr.transform(frag_dataset)
75
76          pred = m.predict(test_dataset)
77          pred = pd.DataFrame(
78              pred,
79              index=test_dataset.ids,
80              columns=["Molecule"]
81          )
82
83          pred_frags = m.predict(frag_dataset)
84          pred_frags = pd.DataFrame(
85              pred_frags,
86              index=frag_dataset.ids,
87              columns=["Fragment"]
88          )
89
90          df = pd.merge(pred_frags, pred, right_index=True, left_index=True)
91          df['Contrib'] = df["Molecule"] - df["Fragment"]
92
93          vis = vis_contribs(test_mol, df)
94          vis.savefig(
95              "pred_results/aquatic_toxicity_qsar_vis.png",
96              bbox_inches='tight'
97          )
98
99  if __name__ == "__main__":
100         main()
```

**Explanation:**
In this example, there are three key points in the instruction: (1) GCN training (lines 28-45), (2) calculating atomic contribution (lines 76-91), and (3) visualizing atomic contribution (lines 10-20). In this case, you can select "Yes" for the first question and move on.

Suppose the given program is not training a GCN at line 33 but, say, a simple feed-forward neural network, you may select "Need Modification" and comment "Line 33" in the follow-up question.

However, if more than three lines of code have errors, please select "No".

**More clarifications:**
(1) The annotated program should be treated as a "reference solution" to the task. As you may have already noticed, these tasks are open-ended and can have multiple valid solutions. So, although the annotated program may import certain classes and packages, we don't want to force the agent to necessarily do the same in the "task instruction." But, if you find the classes and packages helpful, feel free to mention them as "domain knowledge."

(2) The agents will be able to install packages for themselves via pip. For example, if it chooses to use mastml, it should use "pip install mastml" to set itself up. During annotation, we tried to make sure that all packages are distributed via pip so that the agent should be able to install, but there might be a few mistakes. If the program uses something that is not available via pip but is critical to completing the task, please let us know.

## G.3 KNOWLEDGE EXAMPLE PROVIDED TO DOMAIN EXPERTS DURING ANNOTATION

**Task Instruction:** Train a (1) **graph convolutional network** on the given dataset to predict the aquatic toxicity of compounds. Use the resulting model to (2) **compute** and (3) **visualize the atomic contributions** to molecular activity of the given test example compound. Save the figure as "pred_results/aquatic_toxicity_qsar_vis.png".

**Program:**

```
1  import os
2  os.environ["TF_USE_LEGACY_KERAS"] = "1"
3
4  from rdkit import Chem
5  from rdkit.Chem.Draw import SimilarityMaps
6
7  import pandas as pd
8  import deepchem as dc
9
10 ######
11 # (3) This part defines a function for visualizing atomic contributions. One relevant
         piece of domain knowledge you might want to provide to the AI agent or your
         junior student working on this task is about how to draw atomic contributions (
         with rdkit), e.g., by mentioning the required functions.
12
13 def vis_contribs(mol, df, smi_or_sdf = "sdf"):
14     wt = {}
15     if smi_or_sdf == "smi":
16         for n,atom in enumerate(
17             Chem.rdmolfiles.CanonicalRankAtoms(mol)
18         ):
19             wt[atom] = df.loc[mol.GetProp("_Name"),"Contrib"][n]
20     if smi_or_sdf == "sdf":
21         for n,atom in enumerate(range(mol.GetNumHeavyAtoms())):
22             wt[atom] = df.loc[Chem.MolToSmiles(mol),"Contrib"][n]
23     return SimilarityMaps.GetSimilarityMapFromWeights(mol,wt)
24
25 ######
26
27 def main():
28     DATASET_FILE = os.path.join(
29         'benchmark/datasets/aquatic_toxicity',
30         'Tetrahymena_pyriformis_OCHEM.sdf'
31     )
32
33     ######
34     # (1) This part loads the data and trains a GCN. One relevant piece of domain
          knowledge you might want to provide to the AI agent or your junior student
          working on this task is about what IGC50 means and why that column is the gold
          label for aquatic toxicity.
35
36     mols = [
37         m
38         for m in Chem.SDMolSupplier(DATASET_FILE)
39         if m is not None
40     ]
41     loader = dc.data.SDFLoader(
42         tasks=["IGC50"],
43         featurizer=dc.feat.ConvMolFeaturizer(),
44         sanitize=True
45     )
46     dataset = loader.create_dataset(DATASET_FILE, shard_size=5000)
47
48     m = dc.models.GraphConvModel(
49         1,
50         mode="regression",
51         batch_normalize=False
52     )
53     m.fit(dataset, nb_epoch=40)
54
55     ######
56
57     TEST_DATASET_FILE = os.path.join(
58         'benchmark/datasets/aquatic_toxicity',
59         'Tetrahymena_pyriformis_OCHEM_test_ex.sdf'
60     )
61     test_mol = [
62         m
63         for m in Chem.SDMolSupplier(TEST_DATASET_FILE)
64         if m is not None
```

```
 65      ][0]
 66      test_dataset = loader.create_dataset(
 67          TEST_DATASET_FILE,
 68          shard_size=5000
 69      )
 70
 71      loader = dc.data.SDFLoader(
 72          tasks=[],
 73          featurizer=dc.feat.ConvMolFeaturizer(
 74              per_atom_fragmentation=True
 75          ),
 76          sanitize=True
 77      )
 78      frag_dataset = loader.create_dataset(
 79          TEST_DATASET_FILE,
 80          shard_size=5000
 81      )
 82
 83      tr = dc.trans.FlatteningTransformer(frag_dataset)
 84      frag_dataset = tr.transform(frag_dataset)
 85
 86      ######
 87      # (2) This part uses the trained GCN to predict the test example's toxicity and
           calculate the atomic contributions. One relevant piece of domain knowledge you
           might want to provide to the AI agent or your junior student working on this
           task is about how atomic contributions may be calculated, i.e. predicting the
           toxicity of the complete compound and those of compound fragments (with one atom
            removed), then making a subtraction to find the contribution of the removed
           atom.
 88
 89      pred = m.predict(test_dataset)
 90      pred = pd.DataFrame(
 91          pred,
 92          index=test_dataset.ids,
 93          columns=["Molecule"]
 94      )
 95
 96      pred_frags = m.predict(frag_dataset)
 97      pred_frags = pd.DataFrame(
 98          pred_frags,
 99          index=frag_dataset.ids,
100          columns=["Fragment"]
101      )
102
103      df = pd.merge(pred_frags, pred, right_index=True, left_index=True)
104      df['Contrib'] = df["Molecule"] - df["Fragment"]
105
106      ######
107
108      vis = vis_contribs(test_mol, df)
109      vis.savefig(
110          "pred_results/aquatic_toxicity_qsar_vis.png",
111          bbox_inches='tight'
112      )
113
114  if __name__ == "__main__":
115      main()
```

# H  RUBRIC EXAMPLES

In this section, we show two rubrics generated by GPT-4o (Listing H.1, H.3) and their final versions revised by domain experts (Listing H.2, H.4).

Listing H.1: An example rubric of a Computational Chemistry task generated by GPT-4o without expert revision.

```
1  {
2    "data_loading": [
3      {
4        "name": "Initialize Data Loader for Training",
5        "description": "Successfully initializes the MyClintoxLoader object for
         training data with correct parameters: featurizer='ECFP', tasks=['FDA_APPROVED',
         'CT_TOX'], feature_field='smiles', and correct file path 'benchmark/datasets/
         clintox/clintox_train.csv'.",
6        "points": 5
7      },
8      {
9        "name": "Load Training Dataset",
10       "description": "Successfully loads the training dataset using the train_loader
         object and correctly assigns the dataset to train_dataset.",
11       "points": 5
12     },
13     {
14       "name": "Initialize Data Loader for Testing",
15       "description": "Successfully initializes the MyClintoxLoader object for test
         data with correct parameters: featurizer='ECFP', tasks=['FDA_APPROVED', 'CT_TOX
         '], feature_field='smiles', and correct file path 'benchmark/datasets/clintox/
         clintox_test.csv'.",
16       "points": 5
17     },
18     {
19       "name": "Load Testing Dataset",
20       "description": "Successfully loads the test dataset using the test_loader
         object and correctly assigns the dataset to test_dataset.",
21       "points": 5
22     }
23   ],
24   "data_processing": [
25     {
26       "name": "Transform Data",
27       "description": "Applies required transformations to the training dataset using
         the specified transformers (e.g., 'balancing').",
28       "points": 5
29     }
30   ],
31   "modeling_or_analysis_or_visualization": [
32     {
33       "name": "Initialize Model",
34       "description": "Successfully initializes the MultitaskClassifier with
         parameters: number of tasks equal to length of CLINTOX_TASKS, n_features=1024,
         layer_sizes=[1000], dropouts=[0.25], learning_rate=0.001, and batch_size=50.",
35       "points": 10
36     },
37     {
38       "name": "Fit Model",
39       "description": "Successfully fits the model using the train_dataset.",
40       "points": 10
41     },
42     {
43       "name": "Predict Using Model",
44       "description": "Successfully uses the trained model to predict scores on the
         test_dataset, correctly applying any necessary test_transformers.",
45       "points": 10
46     }
47   ],
48   "output_formatting": [
49     {
50       "name": "Format Output DataFrame",
51       "description": "Creates a pandas DataFrame named test_scores_df containing '
         smiles', 'FDA_APPROVED', and 'CT_TOX' columns with correctly assigned test
         scores.",
52       "points": 5
53     }
54   ],
55   "output_saving": [
56     {
57       "name": "Save Predictions to CSV",
```

```
58          "description": "Correctly saves the test_scores_df to a CSV file at '
              pred_results/clintox_test_pred.csv' without an index.",
59          "points": 5
60        }
61    ],
62    "total_points": 65
63 }
```

Listing H.2: An example rubric revised by an expert by adding the available points to two items.

```
 1 {
 2    "data_loading": [
 3      {
 4        "name": "Initialize Data Loader for Training",
 5        "description": "Successfully initializes the MyClintoxLoader object for
              training data with correct parameters: featurizer='ECFP', tasks=['FDA_APPROVED',
              'CT_TOX'], feature_field='smiles', and correct file path 'benchmark/datasets/
              clintox/clintox_train.csv'.",
 6        "points": 10
 7      },
 8      {
 9        "name": "Load Training Dataset",
10        "description": "Successfully loads the training dataset using the train_loader
              object and correctly assigns the dataset to train_dataset.",
11        "points": 5
12      },
13      {
14        "name": "Initialize Data Loader for Testing",
15        "description": "Successfully initializes the MyClintoxLoader object for test
              data with correct parameters: featurizer='ECFP', tasks=['FDA_APPROVED', 'CT_TOX
              '], feature_field='smiles', and correct file path 'benchmark/datasets/clintox/
              clintox_test.csv'.",
16        "points": 5
17      },
18      {
19        "name": "Load Testing Dataset",
20        "description": "Successfully loads the test dataset using the test_loader
              object and correctly assigns the dataset to test_dataset.",
21        "points": 5
22      }
23    ],
24    "data_processing": [
25      {
26        "name": "Transform Data",
27        "description": "Applies required transformations to the training dataset using
              the specified transformers (e.g., 'balancing').",
28        "points": 5
29      }
30    ],
31    "modeling_or_analysis_or_visualization": [
32      {
33        "name": "Initialize Model",
34        "description": "Successfully initializes the MultitaskClassifier with
              parameters: number of tasks equal to length of CLINTOX_TASKS, n_features=1024,
              layer_sizes=[1000], dropouts=[0.25], learning_rate=0.001, and batch_size=50.",
35        "points": 15
36      },
37      {
38        "name": "Fit Model",
39        "description": "Successfully fits the model using the train_dataset.",
40        "points": 10
41      },
42      {
43        "name": "Predict Using Model",
44        "description": "Successfully uses the trained model to predict scores on the
              test_dataset, correctly applying any necessary test_transformers.",
45        "points": 10
46      }
47    ],
48    "output_formatting": [
49      {
50        "name": "Format Output DataFrame",
51        "description": "Creates a pandas DataFrame named test_scores_df containing '
              smiles', 'FDA_APPROVED', and 'CT_TOX' columns with correctly assigned test
              scores.",
52        "points": 5
53      }
54    ],
55    "output_saving": [
56      {
```

```
57        "name": "Save Predictions to CSV",
58        "description": "Correctly saves the test_scores_df to a CSV file at '
       pred_results/clintox_test_pred.csv' without an index.",
59        "points": 5
60      }
61    ],
62    "total_points": 75
63  }
```

Listing H.3: An example rubric of a Geographical Information Science task generated by GPT-4o without expert revision.

```
1  {
2    "data_loading": [
3      {
4        "name": "Load Bathymetry Data",
5        "description": "Correctly loads the bathymetry raster data from the path '
         benchmark/datasets/CoralSponge/CatalinaBathymetry.tif'.",
6        "points": 10
7      },
8      {
9        "name": "Load Coral and Sponge Data",
10       "description": "Correctly reads the coral and sponge data from the path '
         benchmark/datasets/CoralSponge/CoralandSpongeCatalina.geojson'.",
11       "points": 10
12     },
13     {
14       "name": "CRS Transformation",
15       "description": "Correctly transforms the CRS of the GeoDataFrame to EPSG
         :4326.",
16       "points": 5
17     }
18   ],
19   "data_processing": [
20     {
21       "name": "Elevation Conversion",
22       "description": "Correctly converts elevation values by multiplying with -1.",
23       "points": 10
24     },
25     {
26       "name": "Calculate Gradient",
27       "description": "Accurately calculates the gradient (grad_x, grad_y) using numpy
         's gradient function.",
28       "points": 10
29     },
30     {
31       "name": "Calculate Slope",
32       "description": "Correctly calculates the slope in degrees from the gradients.",
33       "points": 10
34     },
35     {
36       "name": "Calculate Aspect",
37       "description": "Correctly calculates the aspect in degrees and adjusts any
         negative values.",
38       "points": 10
39     },
40     {
41       "name": "Coordinate to Raster Index Conversion",
42       "description": "Correctly implements the function to convert coordinates to
         raster grid indices.",
43       "points": 5
44     },
45     {
46       "name": "Extract Slope and Aspect",
47       "description": "Extracts slope and aspect values for each point in the
         GeoDataFrame correctly.",
48       "points": 10
49     },
50     {
51       "name": "Add Slope and Aspect to GeoDataFrame",
52       "description": "Successfully adds the extracted slope and aspect values as new
         columns to the GeoDataFrame.",
53       "points": 5
54     },
55     {
56       "name": "Group by VernacularNameCategory",
57       "description": "Correctly groups the GeoDataFrame by 'VernacularNameCategory'
         and computes mean values for slope and aspect.",
58       "points": 5
59     }
60   ],
61   "modeling_or_analysis_or_visualization": [
62     {
63       "name": "Bar Plot for Mean Slope",
64       "description": "Correctly creates a bar plot showing the mean slope per species
         .",
65       "points": 10
66     },
67     {
68       "name": "Bar Plot for Mean Aspect",
```

```
69        "description": "Correctly creates a bar plot showing the mean aspect per
      species.",
70        "points": 10
71      }
72    ],
73    "output_formatting": [
74      {
75        "name": "Plot Descriptions",
76        "description": "Properly sets plot titles, axis labels, and ensures x-ticks are
      rotated for readability.",
77        "points": 5
78      }
79    ],
80    "output_saving": [
81      {
82        "name": "Save Plots",
83        "description": "Saves the plots as 'mean_slope_per_species.png', '
      mean_aspect_per_species.png', and 'pred_results/CoralandSponge.png'.",
84        "points": 5
85      }
86    ],
87    "total_points": 120
88 }
```

Listing H.4: An example rubric of a Geographical Information Science task revised by an expert by reducing the available points for several items.

```
1  {
2    "data_loading": [
3      {
4        "name": "Load Bathymetry Data",
5        "description": "Correctly loads the bathymetry raster data from the path '
      benchmark/datasets/CoralSponge/CatalinaBathymetry.tif'.",
6        "points": 5
7      },
8      {
9        "name": "Load Coral and Sponge Data",
10       "description": "Correctly reads the coral and sponge data from the path '
      benchmark/datasets/CoralSponge/CoralandSpongeCatalina.geojson'.",
11       "points": 5
12     },
13     {
14       "name": "CRS Transformation",
15       "description": "Correctly transforms the CRS of the GeoDataFrame to EPSG
      :4326.",
16       "points": 5
17     }
18   ],
19   "data_processing": [
20     {
21       "name": "Elevation Conversion",
22       "description": "Correctly converts elevation values by multiplying with -1.",
23       "points": 5
24     },
25     {
26       "name": "Calculate Gradient",
27       "description": "Accurately calculates the gradient (grad_x, grad_y) using numpy
      's gradient function.",
28       "points": 5
29     },
30     {
31       "name": "Calculate Slope",
32       "description": "Correctly calculates the slope in degrees from the gradients.",
33       "points": 10
34     },
35     {
36       "name": "Calculate Aspect",
37       "description": "Correctly calculates the aspect in degrees and adjusts any
      negative values.",
38       "points": 10
39     },
40     {
41       "name": "Coordinate to Raster Index Conversion",
42       "description": "Correctly implements the function to convert coordinates to
      raster grid indices.",
43       "points": 5
44     },
45     {
46       "name": "Extract Slope and Aspect",
```

```
47          "description": "Extracts slope and aspect values for each point in the
        GeoDataFrame correctly.",
48          "points": 10
49        },
50        {
51          "name": "Add Slope and Aspect to GeoDataFrame",
52          "description": "Successfully adds the extracted slope and aspect values as new
        columns to the GeoDataFrame.",
53          "points": 5
54        },
55        {
56          "name": "Group by VernacularNameCategory",
57          "description": "Correctly groups the GeoDataFrame by 'VernacularNameCategory'
        and computes mean values for slope and aspect.",
58          "points": 5
59        }
60      ],
61      "modeling_or_analysis_or_visualization": [
62        {
63          "name": "Bar Plot for Mean Slope",
64          "description": "Correctly creates a bar plot showing the mean slope per species
        .",
65          "points": 10
66        },
67        {
68          "name": "Bar Plot for Mean Aspect",
69          "description": "Correctly creates a bar plot showing the mean aspect per
        species.",
70          "points": 10
71        }
72      ],
73      "output_formatting": [
74        {
75          "name": "Plot Descriptions",
76          "description": "Properly sets plot titles, axis labels, and ensures x-ticks are
        rotated for readability.",
77          "points": 5
78        }
79      ],
80      "output_saving": [
81        {
82          "name": "Save Plots",
83          "description": "Saves the plots as 'mean_slope_per_species.png', '
        mean_aspect_per_species.png', and 'pred_results/CoralandSponge.png'.",
84          "points": 5
85        }
86      ],
87      "total_points": 100
88 }
```

# I PROMPT TEMPLATES

In this section, we document the templates used to prompt LLMs for different frameworks (Section 3): direct prompting (Table I.1), self-debug (Table I.2), and OpenHands CodeAct (Table I.3).

Table I.1: Prompt template for direct prompting (Section 3). `domain_knowledge` is optional.

> You are an expert Python programming assistant that helps scientist users to write high-quality code to solve their tasks.
> Given a user request, you are expected to write a complete program that accomplishes the requested task and save any outputs in the correct format.
> Please wrap your program in a code block that specifies the script type, python. For example:
> ```python
> print(``Hello World!'')
> ```
>
> Please keep your response concise and do not use a code block if it's not intended to be executed.
> Please do not suggest a few line changes, incomplete program outline, or partial code that requires the user to modify.
> Please do not use any interactive Python commands in your program, such as `!pip install numpy`, which will cause execution errors.
>
> Here's the user request you need to work on:
> {task_instruction}
> {domain_knowledge}
> You can access the dataset at `{dataset_path}`. Here is the directory structure of the dataset:
> ```
> {dataset_folder_tree}
> ```
> Here are some helpful previews for the dataset file(s):
> {datase_preview}

Table I.2: Prompt template for self-debug (Section 3). `domain_knowledge` is optional.

> You are an expert Python programming assistant that helps scientist users to write high-quality code to solve their tasks.
> Given a user request, you are expected to write a complete program that accomplishes the requested task and save any outputs in the correct format.
> Please wrap your program in a code block that specifies the script type, python. For example:
> ```python
> print(``Hello World!'')
> ```
>
> The user may execute your code and report any exceptions and error messages.
> Please address the reported issues and respond with a fixed, complete program.
>
> Please keep your response concise and do not use a code block if it's not intended to be executed.
> Please do not suggest a few line changes, incomplete program outline, or partial code that requires the user to modify.
> Please do not use any interactive Python commands in your program, such as `!pip install numpy`, which will cause execution errors.
>
> Here's the user request you need to work on:
> {task_instruction}
> {domain_knowledge}
> You can access the dataset at `{dataset_path}`. Here is the directory structure of the dataset:
> ```
> {dataset_folder_tree}
> ```
> Here are some helpful previews for the dataset file(s):
> {datase_preview}

Table I.3: Prompt template for OpenHands CodeAct (Section 3). `domain_knowledge` is optional.

---

You are an expert Python programming assistant that helps scientist users to write high-quality code to solve their tasks.
Given a user request, you are expected to write a complete program that accomplishes the requested task and save any outputs to `/workspace/pred_results/` in the correct format.

Here's the user request you need to work on:
{task_instruction}
{domain_knowledge}
You can access the dataset at `{dataset_path}`. Here is the directory structure of the dataset:
```
{dataset_folder_tree}
```
Here are some helpful previews for the dataset file(s):
{datase_preview}

Please save your program as `/workspace/pred_programs/{pred_program_name}`.
Then, please run the program to check and fix any errors.
Please do NOT run the program in the background.
If the program uses some packages that are incompatible, please figure out alternative implementations and do NOT restart the environment.

---

## J  PUBLICATIONS, REPOSITORIES, AND LICENSES

In this section, we list all referred publications (Table J.1, J.2) and repositories (Table J.3) during data collection (Section 2.2). We also include the repositories' licenses in Table J.3, J.4, and J.5.

Table J.1: List of Bioinformatics and Computational Chemistry publications referred to during data collection (Section 2.2).

| Domain | Title | Citation |
|---|---|---|
| Bioinfomatics | Automated Inference of Chemical Discriminants of Biological Activity | Raschka et al. (2018) |
| | CellProfiler: image analysis software for identifying and quantifying cell phenotypes | Carpenter et al. (2006) |
| | DeepPurpose: A Deep Learning Library for Drug-Target Interaction Prediction | Huang et al. (2020) |
| | ADMET-AI: a machine learning ADMET platform for evaluation of large-scale chemical libraries | Swanson et al. (2024) |
| | Prediction and mechanistic analysis of drug-induced liver injury (DILI) based on chemical structure | Liu et al. (2021) |
| | SCANPY: large-scale single-cell gene expression data analysis | Wolf et al. (2018) |
| | A Python library for probabilistic analysis of single-cell omics data | Gayoso et al. (2022) |
| | MUON: multimodal omics analysis framework | Bredikhin et al. (2022) |
| | Scirpy: a Scanpy extension for analyzing single-cell T-cell receptor-sequencing data | Sturm et al. (2020) |
| | The scverse project provides a computational ecosystem for single-cell omics data analysis | Virshup et al. (2023) |
| Computational Chemistry | MoleculeNet: a benchmark for molecular machine learning | Wu et al. (2018) |
| | Accelerating high-throughput virtual screening through molecular pool-based active learning | Graff et al. (2021) |
| | Is Multitask Deep Learning Practical for Pharma? | Ramsundar et al. (2017) |
| | Discovery of a structural class of antibiotics with explainable deep learning | Wong et al. (2024) |
| | Papyrus: a large-scale curated dataset aimed at bioactivity predictions | Béquignon et al. (2023) |
| | ProLIF: a library to encode molecular interactions as fingerprints | Bouysset & Fiorucci (2021) |
| | Python Materials Genomics (pymatgen): A robust, open-source python library for materials analysis | Ong et al. (2013) |
| | Benchmarks for interpretation of QSAR models | Matveieva & Polishchuk (2021) |
| | Matminer: An open source toolkit for materials data mining | Ward et al. (2018) |
| | The Materials Simulation Toolkit for Machine learning (MAST-ML): An automated open source toolkit to accelerate data-driven materials research | Jacobs et al. (2020) |
| | Robust model benchmarking and bias-imbalance in data-driven materials science: a case study on MODNet | De Breuck et al. (2021a) |
| | Materials property prediction for limited datasets enabled by feature selection and joint learning with MODNet | De Breuck et al. (2021b) |
| Bioinfomatics & Computational Chemistry | Deep Learning for the Life Sciences | Ramsundar et al. (2019) |

Table J.2: List of Geographical Information Science and Psychology & Cognitive Neuroscience publications referred to during data collection (Section 2.2).

| Domain | Title | Citation |
|---|---|---|
| Geographical Information Science | eofs: A Library for EOF Analysis of Meteorological, Oceanographic, and Climate Data | Dawson (2016) |
| | The Open Global Glacier Model (OGGM) v1.1 | Maussion et al. (2019) |
| | Human selection of elk behavioural traits in a landscape of fear | Ciuti et al. (2012) |
| | Investigating the preferences of local residents toward a proposed bus network redesign in Chattanooga, Tennessee | Ziedan et al. (2021) |
| | Urban wildlife corridors: Building bridges for wildlife and people | Zellmer & Goto (2022) |
| | Urban climate effects on extreme temperatures in Madison, Wisconsin, USA | Schatz & Kucharik (2015) |
| | Model Animal Home Range | Fleming (2024) |
| | Run geoprocessing tools with Python | Zandbergen (2024) |
| | Model How land subsidence affects flooding | Andeweg & Kuijpers (2024) |
| | Predict deforestation in the Amazon rain forest | ESRI (2024a) |
| | NOAA Deep Sea Corals Research and Technology Program | Hourigan (2023) |
| | Chart coral and sponge distribution factors with Python | Robinson (2023) |
| | Assess access to public transit | ESRI (2024b) |
| | Build a model to connect mountain lion habitat | ESRI (2024c) |
| | Analyze urban heat using kriging | Krause (2024) |
| | Assess burn scars with satellite imagery | ESRI (2024d) |
| Psychology & Cognitive Neuroscience | BioPsyKit: A Python package for the analysis of biopsychological data | Richer et al. (2021) |
| | NeuroKit2: A Python toolbox for neurophysiological signal processing | Makowski et al. (2021) |
| | Modeling Human Syllogistic Reasoning: The Role of "No Valid Conclusion" | Riesterer et al. (2019) |
| | Analyzing the Differences in Human Reasoning via Joint Nonnegative Matrix Factorization | Brand et al. (2020) |
| | Generate your neural signals from mine: individual-to-individual EEG converters | Lu & Golomb (2023) |

Table J.3: List of 31 repositories adapted during data collection (Section 2.2) and their licenses. [†]Adaption allowed for non-commercial use; we include their full licenses as Table J.4 and J.5.

| GitHub Repositories | License |
|---|---|
| deepchem/deepchem
coleygroup/molpal
swansonk14/admet_ai
martin-sicho/papyrus-scaffold-visualizer
OlivierBeq/Papyrus-scripts
mad-lab-fau/BioPsyKit
materialsproject/pymatgen
neuropsychology/NeuroKit
nriesterer/syllogistic-nvc
brand-d/cogsci-jnmf
uw-cmg/MAST-ML
ZitongLu1996/EEG2EEG
ResidentMario/geoplot
ppdebreuck/modnet | MIT |
| geopandas/geopandas
kexinhuang12345/DeepPurpose
felixjwong/antibioticsai
SciTools/iris
OGGM/oggm
scverse/scanpy
scverse/scvi-tools
scverse/muon
scverse/scirpy
GeoStat-Framework/PyKrige | BSD-3-Clause |
| psa-lab/predicting-activity-by-machine-learning
chemosim-lab/ProLIF | Apache-2.0 |
| anikaliu/CAMDA-DILI
ajdawson/eofs | GPL-3.0 |
| Solve-Geosolutions/transform_2022 | CC-BY-3.0-AU |
| rasterio/rasterio
hackingmaterials/matminer | Copyrighted[†] |

Table J.4: License for rasterio/rasterio.

Table J.5: License for hackingmaterials/matminer.

