# OpenReview forum: "ScienceAgentBench: Toward Rigorous Assessment of Language Agents for Data-Driven Scientific Discovery"
_ICLR.cc/2025/Conference — ICLR 2025 Poster_

### Official Review · Reviewer_Ws3K · 2024-10-28

**Soundness:** 3
**Presentation:** 2
**Contribution:** 3
**Rating:** 5
**Confidence:** 5

**Summary:**

This paper introduces a novel benchmark, ScienceAgentBench, designed to assess language agents' performance in data-driven scientific exploration. It meticulously curates 102 diverse tasks sourced from 44 peer-reviewed publications spanning four disciplines (Bio, Chem, Information Sci, Psy & Cog Neuroscience), subsequently validated by nine subject matter experts. Employing a variety of evaluation metrics, the study examines the efficacy of generated programs, their execution outcomes, and associated costs. By evaluating five LLMs, including both open-weight and proprietary models, across three frameworks—direct prompting, OpenHands, and self-debug—the findings underscore the current limitations of language agents in generating code for data-driven discovery.

**Strengths:**

(1) Writing: The clarity of this paper make it well-written and easy to comprehend.

(2) Benchmark: This paper introduces ScienceAgentBench, a framework tailored for assessing language agents in the realm of data-driven scientific exploration. It emphasizes scientific authenticity through collaboration with subject matter experts, establishes rigorous evaluation criteria, and maintains meticulous control over multi-stage quality assurance.

(3) Experiments: The paper evaluates three open-source models and two API-based models, conducting detailed assessments and in-depth analyses to provide comprehensive insights.

**Weaknesses:**

(1) It appears that the emphasis of this paper leans more towards Data Science or data-driven discovery rather than scientific discovery.

(2) Task Annotation in Section 2.2 seems labor-intensive and time-consuming due to the involvement of identifying code, preprocessing data, implementing code, and writing dataset information. Are there any automated annotation or data collection methods available?

(3) How is the ground truth for each task defined and generated? Are there any automated validation methods that could streamline this process instead of relying solely on multiple rounds of manual validation by annotators?

(4) Could you elaborate on how the evaluation criteria outlined in Table 1 were established?

(5) Regarding the validation of generated Python programs during inference and the utilization of CodeBERTScore to assess token-level embeddings, have you considered employing a self-consistency strategy to validate multiple outputs over time?

(6) How is the validity of outputs generated by GPT-4o for the four heterogeneous datasets depicted in Figure 1 verified?

(7) Given the focus on code generation for data science, have you considered evaluating or providing the performance of code generation models like Codellama and DeepSeek-Coder?

(8) There appears to be inconsistency in the citation format, as observed in instances such as line 249 and line 251. Would it be possible to ensure uniformity in citation formatting throughout the paper?

**Questions:**

Please see the Weaknesses.

**Details Of Ethics Concerns:**

A significant area of concern regarding this research pertains to the safety implications associated with the proposed language agent. Particularly in fields like bio or chemistry, there is a potential risk that the language agent could inadvertently synthesize toxic or dangerous chemicals. It is recommended that the authors address these safety considerations by conducting a thorough analysis of the potential risks associated with the language agent presented in this study.

---

> ### Author Response · Authors · 2024-11-18
> **Author Response to Reviewer Ws3K (Part 1/4: Clarifications)**
>
> Thanks to Reviewer Ws3K for recognizing ScienceAgentBench as a benchmark that “emphasizes scientific authenticity,” “establishes rigorous evaluation criteria,” and “maintains meticulous control” over data quality. In the first two posts, we clarify some concerns and elaborate on some questions mentioned by the reviewer. In the third post, we would like to emphasize our manual efforts in developing this benchmark and respectfully disagree with Reviewer Ws3K on three comments listed as weaknesses of our submission. In the last post, we provide our thoughts on the safety issues of language agents for scientific discovery.
>
> ### [W1] It appears that the emphasis of this paper leans more towards Data Science or data-driven discovery rather than scientific discovery.
> ***
>
> We appreciate it if the reviewer can elaborate on why focusing on data science or data-driven discovery is a weakness of our submission. As stated in the title, our benchmark is developed to rigorously assess "language agents for data-driven scientific discovery." Also, we would like to clarify that data science and data-driven discovery are **important paradigms in scientific discovery** but not independent concepts as the reviewer seems to suggest. Data-driven discovery, or data science, has been recognized as "a new, fourth paradigm for scientific exploration" since 2009 [1]. Scientists have been interested in deriving new insights from big data, but they are overwhelmed by the amount of data and often lack programming skills to analyze them [2]. A language agent that can automate tasks in data-driven discovery would help them save hours of effort.
>
> ### [W4] Could you elaborate on how the evaluation criteria outlined in Table 1 were established?
> ***
>
> The evaluation criteria in our benchmark are tailored to each task and established by measuring whether an LLM-generated program accurately reproduces the result of the annotated program. Since the annotated programs are adapted from open-source repositories of peer-reviewed publications and validated by subject matter experts, their execution results faithfully represent part of the research outcomes in those publications. An agent that is capable of implementing a program correctly to reproduce the result would also produce a correct program for similar tasks in real-world scenarios.
>
> For example, we have executed our annotated program to train a multitask model on the Clintox dataset for five independent runs and consistently observe that the model achieves at least 0.77 ROC-AUC score on the test set. Thus, we use 0.77 as the performance threshold in this evaluation criterion and require the agent to train a model with the same level of performance to be considered successfully completing the task. Evaluation criteria for other tasks are also established following the same principle of reproducing some data-driven discovery results.
>
> We will add the above clarification to our paper.
>
> ### References
> [1] Tony Hey, et al. The Fourth Paradigm: Data-Intensive Scientific Discovery. Microsoft Research, October 2009. ISBN 978-0-98254420-4.
>
> [2] Gordon Bell, et al. Beyond the data deluge. Science, 323(5919):1297–1298, 2009. doi: 10.1126/science.1170411

---

> > ### Author Response · Authors · 2024-11-18
> > **Author Response to Reviewer Ws3K (Part 2/4: Clarifications)**
> >
> > ### [W5] Regarding the validation of generated Python programs during inference and the utilization of CodeBERTScore to assess token-level embeddings, have you considered employing a self-consistency strategy to validate multiple outputs over time?
> > ***
> >
> > We agree with the reviewer that self-consistency can be used to improve the LLMs and agent. However, we note that applying self-consistency or majority voting strategies for the complex program outputs in ScienceAgentBench is nontrivial: It is very likely that the programs sampled for a task have unique execution results, so their voting count would all be 1. In such occasions, self-consistency would fail to decide which program to choose. This is unlike math word problems where there is one single answer to each question and such strategies are often applied. Besides, OpenHands CodeAct does not support additionally self-consistency check during inference time due to its encapsulation. To fairly compare the performance and costs of different frameworks, we choose not to add self-consistency to only two of them in this work. Future studies may design self-consistency mechanisms for complex programs and incorporate it as part of their design to achieve better performance on ScienceAgentBench. Finally, we note that CodeBERTScore is only used for evaluation purposes and not related to inference-time validation.
> >
> > ### [W6] How is the validity of outputs generated by GPT-4o for the four heterogeneous datasets depicted in Figure 1 verified?
> > ***
> >
> > This question seems ambiguous and we would appreciate further clarifications from the reviewer. But here let us clarify three potential misunderstandings: (1) We unify the target output for every task as a self-contained Python program, which means all LLMs/agents are only generating programs but not figures. The generated programs are then executed to produce any figures, if required by a task. The evaluation criteria for the generated programs is clarified in our response to [W4]. (2) For all tasks with figures as outputs, we use GPT-4o as an LLM judge of the figure quality (lines 249--253) following related work, which demonstrates reasonable correlations with human raters. (3) The data visualizations in Figure 1 are produced by the authors using domain-specific tools instead of GPT-4o, and serve as four representative examples to show the heterogeneity of the datasets used for tasks in our benchmark.
> >
> > ### [W7] Given the focus on code generation for data science, have you considered evaluating or providing the performance of code generation models like Codellama and DeepSeek-Coder?
> > ***
> >
> > We select the five general-purpose LLMs to evaluate based on two considerations: (1) They can be flexibly incorporated into different agent frameworks like OpenHands CodeAct, which have an extensive list of natural language instructions as well as non-coding components like web browsing. (2) Most LLMs specialized in code generation, such as the Codellama series, performs similarly or worse than our five selected models on standardized code generation benchmarks, e.g., HumanEval and MBPP [1]. These LLMs also have limited context windows of 16K or 32K tokens, which does not meet our needs, as some of our tasks' input information already has 32K tokens. The only exception is DeepSeek-Coder v2, which has a 128K context window, but our select models perform similarly or better on standardized code generation benchmarks as well. As a result, we choose not to evaluate these LLMs of code in this work. However, we encourage future research to develop better agents using these models and evaluate them on our benchmark.
> >
> > ### References
> > [1] https://mistral.ai/news/mistral-large-2407/

---

> > > ### Author Response · Authors · 2024-11-18
> > > **Author Response to Reviewer Ws3K (Part 3/4: Manual Efforts and Disagreements)**
> > >
> > > ### [W2] Task Annotation in Section 2.2 seems labor-intensive and time-consuming due to the involvement of identifying code, preprocessing data, implementing code, and writing dataset information. Are there any automated annotation or data collection methods available?
> > > ***
> > >
> > > We respectfully disagree with Reviewer Ws3K that "labor-intensive and time-consuming" makes our task annotation process a weakness. To answer the question first, there are no reliable automated annotation or data collection methods to collect high-quality data-driven discovery tasks, to our best knowledge. Even for benchmarks collected automatically, we argue that intensive labor to verify the data is necessary to establish evaluation quality, e.g., SWE-bench Verified [1]. We would also like to kindly remind the reviewer that many highly impactful datasets and benchmarks have been built with intensive labor over long time periods for AI development, such as ImageNet and Penn Treebank. As acknowledged by Reviewer fdqA, one of the most important contributions of this work is our "dedication and thoroughness" to conduct "extensive data curation and human annotation." With an average of 2.5--3 hours spent for annotating each task, we have invested 250--300 person-hours to merely collect the benchmark, not to mention additional validations conducted by the annotators and subject matter experts. We believe our efforts make our "labor-intensive and time-consuming" task annotation a strength of this work.
> > >
> > > ### [W3] How is the ground truth for each task defined and generated? Are there any automated validation methods that could streamline this process instead of relying solely on multiple rounds of manual validation by annotators?
> > > ***
> > >
> > > The ground truth program for each task is first extracted as is, instead of written by humans or generated by any models, from the open-source repositories of peer-reviewed publications to ensure their scientific authenticity. Then, our annotators make necessary modifications to remove redundant lines and load the datasets in our benchmark. Finally, the ground truth programs are validated by subject matter experts, as well as other annotators.
> > >
> > > Similar to our response to W2, we do not recognize any suitable automated validation methods in existing literature. Even if such a method exists, we believe that it cannot simply replace our multi-round data validation: One key design principle of ScienceAgentBench (lines 079--085) is to ensure the authenticity of collected tasks through co-design with subject matter experts, who are irreplaceable by automated methods.
> > >
> > > ### [W8] Inconsistency in the citation format.
> > > ***
> > > Our different citation formats are not inconsistent but strictly adhere to the APA in-text citation standard, where `\citet` should be used if the in-text citation serves as a noun in a sentence. We kindly refer the reviewer to the Purdue OWL citation guide for more details [2].
> > >
> > > ### References
> > > [1] https://openai.com/index/introducing-swe-bench-verified/
> > >
> > > [2] https://owl.purdue.edu/owl/research_and_citation/apa_style/apa_formatting_and_style_guide/in_text_citations_author_authors.html

---

> > > > ### Author Response · Authors · 2024-11-18
> > > > **Author Response to Reviewer Ws3K (Part 4/4: Safety and Ethics)**
> > > >
> > > > We agree with Reviewer Ws3K that the safety implications of language agents for data-driven scientific discovery are important. We appreciate that the reviewer mentions this issue. However, we would like to first clarify that our work is a benchmark paper and does not train or develop any new language agent. Instead, our work proposes ScienceAgentBench and uses this new benchmark to evaluate and analyze **existing LLMs and agents developed by other researchers**. Thus, we are not introducing any safety harms or ethical issues in this paper, and the reviewer's concern about "the proposed language agent" in our research may not be grounded.
> > > >
> > > > Yet, we have discussed with our subject matter experts in Bioinformatics and Computational Chemistry about the risk of synthesizing "toxic or dangerous chemicals." Our thoughts are as follows:
> > > > 1. Our Bioinformatics and Computational Chemistry tasks focus on property prediction, feature analyses, and molecule visualization, which does not involve synthesis or generation of biological or chemical substances.
> > > > 2. Unlike Coscientist [1], agents evaluated in our submission are not connected to any laboratory hardwares. Thus, it is impossible for these agents to produce any dangerous chemicals or substances on their own. Even if they were to be instructed to write code for chemical synthesis in real-world applications, human intervention is still required to grant the access to laboratories, reagents, and equipment.
> > > > 3. The target outputs for every task in ScienceAgentBench are unified as self-contained Python programs. Therefore, the evaluated agents only generate code for processing, analyzing and visualizing scientific data that is already publicly available. They are not instructed to generate chemical reactions or synthesis pathways.
> > > >
> > > > Finally, we suggest that "a thorough analysis of the potential risks" of language agents for science is an important research topic [2] but out of the scope of this work. As a benchmark paper, we prioritize discussing the ethical and safety considerations about the **data and tasks** involved (Appendix A). Respecting the data ownership and intellectual property, we made our best effort to cite the original papers, list the repositories, and provide their licenses in Appendix H.
> > > >
> > > > To summarize, our submission contributes an evaluation benchmark to assess existing language agents rigorously, which has limited or no risk in "inadvertently synthesizing toxic or dangerous chemicals." We also recommend the developers of these agents to consider such potential risks seriously and provide effective intervention mechanisms for users.
> > > >
> > > > ### References
> > > > [1] Daniil A. Boiko, Robert MacKnight, Ben Kline, and Gabe Gomes. Autonomous chemical research with large language models. Nature, 624:570–578, 2023. doi: https://doi.org/10.1038/
> > > > s41586-023-06792-0
> > > >
> > > > [2] Xiangru Tang, et al. Prioritizing Safeguarding Over Autonomy: Risks of LLM Agents for Science. Arxiv preprint 2024. https://arxiv.org/abs/2402.04247

---

> > > > > ### Author Response · Authors · 2024-11-22
> > > > > **Gentle Reminder from Authors**
> > > > >
> > > > > Dear Reviewer Ws3K,
> > > > >
> > > > >
> > > > > As the end of discussion period is approaching, we would like to gently remind you of our responses to your comments. We wonder whether your concerns have been addressed and appreciate any further questions or comments you might have.
> > > > >
> > > > > Sincerely,
> > > > >
> > > > > Authors of Submission12844

---

### Official Review · Reviewer_fdqA · 2024-11-03

**Soundness:** 3
**Presentation:** 3
**Contribution:** 3
**Rating:** 8
**Confidence:** 4

**Summary:**

The ScienceAgentBench framework is introduced in this paper to assess the data-driven scientific discovery capabilities of LLM models. The framework offers both end-to-end and fine-grained metrics in evaluations. Significant room for improvement in scientific tasks was confirmed by implementing the benchmark on various sota models. The benchmark has the potential to serve as a long-term progress indicator for LLM models on scientific reasoning capabilities.

**Strengths:**

The study involved extensive data curation and human annotation, demonstrating the authors' dedication and thoroughness. The inclusion of both end-to-end and fine-grained metrics allows for a comprehensive evaluation of models, particularly when the models can only partially solve a problem. Additionally, the exploration and discussion of various interaction methods with the local environment provides valuable insights.

**Weaknesses:**

Coding generation-related tasks may not be representative of some other scientific domains. While recent research has focused on such tasks, the authors could briefly acknowledge this limitations, especially since the benchmark's name suggests a more comprehensive evaluation of broader scientific capabilities.

**Questions:**

Why was VER chosen over CBS when ranking models? High VER but low CBS could still indicate good context understanding, though poor execution. Was it considered to use heuristics / weighted sum to combine all metrics in the final evaluation?

Will setting CBS to 1.0 when SR is 1 introduce bias into the metric? Some argue that this specific treatment can skew the metric's results. While CBS may not be ideal when the model employs a different approach than annotation but still arrives at the correct answer, setting it to 1.0 could lead to inconsistent score interpretations. Additionally, if the ranking is order-based, this specific treatment might not have a significant impact.

---

> ### Author Response · Authors · 2024-11-18
> **Author Response to Reviewer fdqA**
>
> We sincerely appreciate Reviewer fdqA's recognition of our contributions and acknowledgement of our "dedication and thoroughness" to conduct "extensive data curation and human annotation." Our responses to the reviewer's remaining concerns and questions are as follows:
>
> ### [W1] Coding generation-related tasks may not be representative of some other scientific domains.
> ***
>
> We agree with Reviewer fdqA that while code generation is necessary for data-driven scientific discovery, it may not be representative of some other scientific domains. In fact, we have a dedicated section in **Appendix A** to discuss this limitation of our work, and we will revise our manuscript to mention it in the main text. To summarize our discussion in Appendix A, we encourage future studies to carefully examine the agents’ other capabilities that can help with scientific discovery, such as summarizing literature, suggesting ideas, or planning experiments. We focus on code generation because it is relatively easy to verify using well-established automatic metrics, and the resulting program can be directly usable by a scientist without additional efforts to modify or implement. In this way, we can rigorously assess the abilities and limitations of existing agents and minimize the generalization gap for agents developed on our benchmark to real-world scenarios. Due to data-driven scientific discovery and AI, coding is also becoming an increasingly important part of scientists’ workflow. Reliably automating scientific coding tasks has substantial tangible values on its own.
>
> ### [Q1] Why was VER chosen over CBS when ranking models? High VER but low CBS could still indicate good context understanding, though poor execution. Was it considered to use heuristics / weighted sum to combine all metrics in the final evaluation?
> ***
>
> To first clarify, the models and agents evaluated in our work are ranked by a single metric, SR. We also appreciate the suggestion to "use heuristics / weighted sum to combine all metrics in the final evaluation," which we will seriously consider when maintaining the leaderboard in the future. For our manuscript, the order introduced in lines 320--321 and 349--352 is for **selecting the best run of the same model/agent** from its three attempts. Inspired by the Pass@k metric in general code generation tasks, such as HumanEval, our evaluation design also takes the randomness of LLM generation into consideration. To this end, we conduct three independent runs for each model under each agent framework and use the order to select the best run when calculating the final metrics. We prioritize VER over CBS in this order because a program being executable (VER = 1) is a strict a priori for success (SR = 1), while a successful program can take an approach different from the annotation and have a lower CBS. In this way, we try to demonstrate the best performance each model/agent can achieve, while avoiding cherry-picking the numbers, e.g., by reporting the VER from one run but CBS from another.
>
> ### [Q2] Will setting CBS to 1.0 when SR is 1 introduce bias into the metric? Some argue that this specific treatment can skew the metric's results. While CBS may not be ideal when the model employs a different approach than annotation but still arrives at the correct answer, setting it to 1.0 could lead to inconsistent score interpretations. Additionally, if the ranking is order-based, this specific treatment might not have a significant impact.
> ***
>
> As mentioned in our answer to Q1, the ranking is decided by SR alone, and other metrics like CBS complements SR as more comprehensive assessments of the models and agents. We set CBS to 1.0 when SR is 1 to maintain the ranking consistency between CBS and SR. As the reviewer commented, LLM-generated programs can take different approaches than our annotations, and it makes less sense if one model has higher SR and VER but lower CBS than another model, which we have observed in our preliminary experiments. Additionally, we note the CBS scores are mostly clustered between 0.6 and 0.9, making it hard to distinguish high-quality, correct programs from the rest. Based on these rationales, we believe adding this rule for CBS brings more benefits than harm by better crediting the models and agents that successfully solve more tasks. We acknowledge the potential bias or skewed distribution introduced here and are open to more discussions with Reviewer fdqA to find a better solution.

---

> > ### Comment · Reviewer_fdqA · 2024-11-28
> >
> > Thank you for the detailed explanations and proposed revisions. I appreciate you addressing the questions I raised, and I'll be maintaining the positive rating.

---

### Official Review · Reviewer_c5sH · 2024-11-03

**Soundness:** 3
**Presentation:** 3
**Contribution:** 3
**Rating:** 5
**Confidence:** 3

**Summary:**

This paper introduces ScienceAgentBench, a new benchmark designed to test how well language agents can handle tasks in data-driven scientific discovery. The authors collected 102 tasks from 44 peer-reviewed papers across four scientific fields: Bioinformatics, Computational Chemistry, Geographical Information Science, and Psychology & Cognitive Neuroscience. Each task asks the agents to write self-contained Python programs to perform specific scientific activities like data processing, model development, analysis, and visualization.

To ensure the tasks are authentic and to prevent issues with data contamination, the authors involved experts from the respective fields and modified datasets so agents couldn't rely on memorized code. They evaluated five large language models using three different frameworks: direct prompting, OpenHands, and self-debug. The results showed that the best-performing agent could complete only about one-third of the tasks. This highlights the current limitations of language agents in fully automating data-driven scientific discovery and suggests that more advancements are needed.

**Strengths:**

The paper introduces ScienceAgentBench, a novel benchmark for evaluating language agents in data-driven scientific discovery tasks. By incorporating tasks from four diverse scientific disciplines—Bioinformatics, Computational Chemistry, Geographical Information Science, and Psychology & Cognitive Neuroscience—it creatively applies language agents to new domains, filling a gap where existing benchmarks fall short.

The benchmark is rigorously developed with input from nine subject matter experts, ensuring tasks are authentic and challenging. The authors proactively mitigate data contamination by modifying datasets, enhancing the reliability of their evaluation. They use comprehensive evaluation metrics—including Valid Execution Rate (VER), Success Rate (SR), CodeBERTScore (CBS), and computational costs—to provide a holistic assessment of agent performance.

The paper is well-organized and written, utilizing figures and tables to enhance understanding. The authors provide insightful analyses of experimental results, highlighting why current language agents struggle with these tasks. By releasing all code and data, they promote open science and collaboration, significantly contributing to the advancement of AI in scientific research.

**Weaknesses:**

1. The paper evaluates agents using three frameworks but doesn't justify these choices or explore advanced architectures like ReAct or Toolformer. Without including state-of-the-art frameworks that offer advanced reasoning and tool-use capabilities, the study may not fully assess the agents' potential to handle complex scientific tasks. Incorporating such frameworks could provide deeper insights into their capabilities and limitations.
2. Human evaluators who also participated in data collection may introduce bias due to familiarity with the tasks, affecting the objectivity of the assessments. Additionally, the error analysis lacks depth, as specific failure modes are not thoroughly examined. Involving independent evaluators and conducting a detailed error analysis would improve objectivity and help identify areas where agents struggle.
3. The paper doesn't compare the agents' performance with traditional methods or domain-specific tools, making it difficult to assess their practical utility relative to existing solutions. Including such comparisons would provide valuable context to evaluate the agents' real-world usefulness and guide future improvements.
4. Providing expert domain knowledge doesn't consistently improve agent performance and sometimes even decreases it, suggesting agents struggle to integrate this information effectively. Exploring why agents fail to benefit from expert knowledge could lead to better integration strategies and enhance their overall performance.

**Questions:**

1. Have you considered evaluating state-of-the-art frameworks like ReAct or Toolformer incorporating advanced reasoning and tool-use capabilities? Including these could offer deeper insights into the agents' performance on complex tasks.
2. Since evaluators were also involved in data collection, how did you mitigate potential assessment bias? Would involving independent evaluators improve objectivity?
3. Could you provide a more detailed analysis of the standard failure modes encountered by the agents? Understanding specific errors might help identify areas for improvement.
4. Have you compared the agents' performance with traditional methods or domain-specific tools? Including such comparisons could help assess their practical utility relative to existing solutions.

---

> ### Author Response · Authors · 2024-11-18
> **Author Response to Reviewer c5sH (Part 1/2: Clarifications)**
>
> We are grateful that Reviewer c5sH finds our benchmark “novel” and fills a gap “where existing benchmarks fall short” with “authentic and challenging” tasks. We would like to first clarify some concerns in this post and then follow up on other constructive feedback from the reviewer in the next post.
>
> ### [W1 & Q1] Evaluation of frameworks like ReAct or Toolformer.
> ***
>
> We agree with Reviewer c5sH on “including state-of-the-art frameworks to fully assess the agents' potential to handle complex scientific tasks,” but there might be some misunderstanding here. We suggest that ReAct and Toolformer are **no longer** the state-of-the-art frameworks for language agents. Instead, we have included OpenHands CodeAct published in July 2024, which is one of the best open-source frameworks that incorporates both ReAct-style reasoning and Toolformer-like tool-use capabilities. We kindly refer the reviewer to the original papers [1][2] for more details about this framework. Thus, our paper indeed evaluates agents with a state-of-the-art framework, i.e. OpenHands CodeAct, and offers an important insight into its limitation: With Claude-3.5-Sonnet, the simpler self-debug can successfully solve 10.8% more tasks than OpenHands CodeAct while costing 17 times less API fees, which resonates with recent findings that agent frameworks should jointly consider costs and performance to maximize their practical utility [3].
>
> ### [W3 & Q4] Compare the agents' performance with traditional methods or domain-specific tools.
> ***
>
> We would appreciate it if Reviewer c5sH can provide more details, e.g., by naming some examples of "traditional methods or domain-specific tools." Here we attempt to clarify this comment based on our understanding: We assume the reviewer is referring to traditional methods or domain-specific tools for solving each scientific task automatically.
>
> However, we want to clarify that this benchmark is for rigorously evaluating **language agents** on data-driven discovery tasks. To this end, we follow existing work in Table 2 and Section 2.4 and compare the LLM-based agents with a conventional approach, directly prompting LLMs. To the best of our knowledge, there is **no** domain-specific code generation tool or any traditional methods other than LLMs that can perform our tasks well. Reliable automated code generation only became possible very recently with LLMs.
>
> Finally, we want to stress on the difficulty of generating complex programs in our benchmark, and LLM-based agents, such as OpenHands CodeAct evaluated in this paper, have established their practical utility in such real-world tasks [4]. An agent that performs well on our benchmark can find several real-world applications, such as helping scientists to replicate papers that do not release open-source code or write programs to try their new research ideas efficiently.
>
> ### [W4] Exploring why agents fail to benefit from expert knowledge could lead to better integration strategies and enhance their overall performance.
> ***
>
> We agree with the reviewer, and in our manuscript (lines 406--423), we have provided two reasons why agents fail to incorporate expert knowledge: (1) Expert-provided knowledge specifies some specific tools that are less familiar to the agents. (2) The agents do not know how to solve some tasks without expert-provided knowledge and would generate some executable but less meaningful programs, e.g., one that produces an empty figure.
>
> We would also like to gently remind Reviewer c5sH that our paper is proposing a new benchmark for rigorously evaluating existing agents for data-driven scientific discovery, and improving existing agents is important but falls beyond the scope of this work. Using our benchmark, we show that existing LLM-based language agents may not effectively incorporate expert-provided knowledge into their problem-solving process. We believe this is not a weakness of our paper, but an important insight derived with our benchmark that poses a new research question to the community for future research.
>
> ### References
> [1] Xingyao Wang, et al. Executable Code Actions Elicit Better LLM Agents. In ICML 2024. https://arxiv.org/abs/2402.01030
>
> [2] Xingyao Wang, et al. OpenHands: An Open Platform for AI Software Developers as Generalist Agents. Arxiv preprint 2024. https://arxiv.org/abs/2407.16741
>
> [3] Sayash Kapoor, et al. AI Agents That Matter. Arxiv preprint 2024. https://arxiv.org/abs/2407.01502
>
> [4] https://x.com/allhands_ai/status/1857089580236714241

---

> > ### Author Response · Authors · 2024-11-18
> > **Author Response to Reviewer c5sH (Part 2/2: Follow-up Discussions)**
> >
> > ### [W2 & Q2] Human evaluators who also participated in data collection may introduce bias
> > ***
> >
> > We thank Reviewer c5sH for this comment on potential bias that may affect the objectivity of the assessment. It would be appreciated if the reviewer can elaborate on why they think so, but let us explain our rationale for involving human evaluators who also participated in data collection: An important evaluation challenge to develop this benchmark is the open-endedness of our tasks. Although we have annotated one program for each task as a reference solution, it is not the **only** solution. Given the same task instruction, an LLM-based agent can take different approaches to write another correct program that is dissimilar to the annotation.
> >
> > To address this challenge, in our human evaluation, we deliberately ask the annotators to serve as the raters again. Due to their familiarity with the tasks and deeper understanding of the programs, they can more accurately recognize whether an LLM-generated program is correct, even though it can appear to be very different from the annotation. If we involve independent evaluators, they may not capture such equivalences accurately and deduct more scores based on superficial differences. This would lead to more false negatives in the human evaluation. Therefore, we have asked the annotators to judge the correctness of these programs. We also note that at evaluation time, they do not know which model or agent has generated the programs, so there would be no bias towards a certain LLM/agent framework.
> >
> > ### [W2 & Q3] Detailed analysis of the standard failure modes encountered by the agents
> > ***
> >
> > We thank the reviewer for this constructive feedback. As suggested, we conduct and will add a new error analysis of agent trajectories. Using Claude-3.5-Sonnet as the base LLM, we sample 50 error trajectories for OpenHands CodeAct and self-debug respectively. From the 100 error trajectories, we find that both agents need **better reasoning and self-verification capabilities** to make sure their executable programs are also semantically correct (29/50 errors for OpenHands CodeAct and 30/50 errors for self-debug). For instance, when having trouble loading the actual scientific data, the agent may write code to simulate some fake data to make the program executable but produce incorrect results. Similarly, when the agent cannot implement something correctly, e.g., a graph convolutional neural network, it may just turn to implementing a simpler feed-forward network, which underfits the complex data and cannot reproduce the desired performance. These executable but functionally incorrect programs need to be better captured and fixed by improving the agents' reasoning and self-verification in future research.
> >
> > The other major issue for both agents is their ability to **install and configure the environments with domain-specific tools correctly**. Our analysis reveals that both the LLM-generated installation commands in OpenHands CodeAct (10/50 are configuration errors) and human-developed packages used in self-debug (9/50 are configuration errors) are not sufficient to set up some domain-specific tools correctly. This finding echoes with concurrent work [1] that environmental setup for scientific tasks remains challenging for language agents. When the environment is not set up correctly, both agents try to get around domain-specific tools in their programs and use simpler ones, such as developing a random forest model with scikit-learn instead of deep learning models in deepchem.
> >
> > Finally, we find that in 23 of the 50 error trajectories, Claude-3.5-Sonnet was struggling with the specialized commands in OpenHands to edit programs correctly (lines 394--396), especially for longer programs. It would fall into loops of repeatedly generating such commands as shown in Appendix D.1. Such behaviors waste quite a few turns on fixing the use of these commands and largely increase the API cost. Future agent research should reconsider the use of such commands and compare closely with some pipeline-based approaches like the Agentless framework [2].
> >
> > ### References
> > [1] Ben Bogin, et al. SUPER: Evaluating Agents on Setting Up and Executing Tasks from Research Repositories. Arxiv preprint 2024. https://arxiv.org/abs/2409.07440
> >
> > [2] Chunqiu Steven Xia, et al. Agentless: Demystifying LLM-based Software Engineering Agents. Arxiv preprint 2024. https://arxiv.org/abs/2409.07440

---

> > > ### Author Response · Authors · 2024-11-22
> > > **Gentle Reminder from Authors**
> > >
> > > Dear Reviewer c5sH,
> > >
> > > As the end of discussion period is approaching, we would like to gently remind you of our responses to your comments. We wonder whether your concerns have been addressed and appreciate any further questions or comments you might have.
> > >
> > > Sincerely,
> > >
> > > Authors of Submission12844

---

> > > > ### Comment · Reviewer_c5sH · 2024-11-22
> > > > **ScienceAgentBench: Toward Rigorous Assessment of Language Agents for Data-Driven Scientific Discovery**
> > > >
> > > > The authors provided a thorough rebuttal, clarifying their use of state-of-the-art frameworks and addressing concerns about evaluation and error analysis. They identified key failure modes and emphasized the benchmark's potential for advancing data-driven scientific discovery. However, the responses did not fully address the concerns regarding comparisons with traditional methods or the broader utility of the benchmark. Therefore, my original grade remains unchanged, as the rebuttal does not significantly alter my overall evaluation.

---

> > > > > ### Author Response · Authors · 2024-11-22
> > > > > **Additional Clarifications for Reviewer c5sH**
> > > > >
> > > > > Thanks for your response! We are glad to see that our rebuttal has clarified **most** of your concerns, especially on the evaluation of frameworks like ReAct or Toolformer. We believe the reviewer agrees with us that OpenHands CodeAct is one of the state-of-the-art frameworks that incorporates both ReAct-style reasoning and Toolformer-like tool-use capabilities.
> > > > >
> > > > > However, we are still wondering what your remaining concerns are regarding "comparisons with traditional methods or the broader utility of the benchmark." To our understanding, "traditional methods or domain-specific tools" to directly generate such code for these scientific disciplines simply do not exist. **We would appreciate it if Reviewer c5sH could help to name one of such methods or tools they have in mind.** Scientists have to manually write the code by themselves or collaborate with some programmers. Reliable automated code generation only became possible very recently with LLMs.
> > > > >
> > > > > Therefore, in this work, we propose ScienceAgentBench to rigorously evaluate language agents on their abilities to assist scientists with coding tasks in their research workflows, such as replicating papers that do not release open-source code or writing programs to try their new research ideas efficiently. This is an important contribution of our benchmark that has broader utility in (1) helping AI researchers to understand and develop better language agents and (2) helping scientists to accelerate their data-driven discovery process.

---

### Author Response · Authors · 2024-11-21
**Manuscript Updated By Authors**

We would like to thank the reviewers again for their constructive feedback. We have revised our paper to reflect some of the suggestions made by the reviewers:

- In the main text, we have added more references to content in Appendix A, C and D.
- We have moved the discussion of our future work from Section 6 to Appendix A. We have also added more discussion on agent safety, especially our rationales why this work introduces limited or no risk in inadvertently synthesizing toxic or dangerous chemicals.
- We have added Appendix C for more details about how we defined the annotated programs and established success criteria in our benchmark construction process.
- We have also included our detailed error analysis of agent trajectories as Appendix D.2.

---

### Author Response · Authors · 2024-11-27
**Summary of Author Responses and Gentle Reminder of Discussion**

Dear Reviewers,

Since the author-reviewer discussion period is extended for one week, we would like to gently remind you again of our responses and appreciate any acknowledgements or follow-up discussions. Here, we summarize the contributions of our work and our responses to each reviewer:

### Contributions
***

1. We present ScienceAgentBench, a rigorously developed benchmark for evaluating language agents on data-driven scientific discovery tasks. We involve extensive human annotation efforts and include nine subject-matter experts to ensure data quality and scientific authenticity, as well as proactively mitigating data contamination risks.
2. We comprehensively evaluate existing state-of-the-art LLMs and agents with different metrics, provide insightful analysis of our experimental results, and point out potential future directions in developing and evaluating agents for scientific discovery.

### Response to Reviewer c5sH
***

We have reached an agreement with the reviewer that OpenHands CodeAct evaluated in this work is one of the state-of-the-art frameworks, which incorporates both ReAct-style reasoning and Toolformer-like tool-use capabilities. Additionally, we provide a detailed analysis of agent trajectories and clarify that better expert knowledge integration is out of the scope of this benchmark paper. Since we have addressed most of the reviewer's concerns, we appreciate it if the reviewer could adjust their assessment accordingly.

We encourage the reviewer to follow up on our discussion and name one of the "traditional methods or domain-specific tools" they have in mind. To our understanding, methods or tools for reliable code generation simply do not exist other than LLMs.

### Response to Reviewer fdqA
***

We clarify our consensus with Reviewer fdqA that while code generation is necessary for data-driven scientific discovery, it may not be representative of some other scientific domains. We kindly refer the reviewer to Appendix A for relevant discussions on this limitation and why we focus on code generation for data-driven discovery. Besides, we provide detailed responses to Reviewer fdqA's other questions.

### Response to Reviewer Ws3K
***

We kindly point out that the extensive time, labor, and multi-round validation efforts in this work are important contributions rather than weaknesses. To our knowledge, methods to automate the annotation or validation process do not exist (not to mention replacing humans, especially subject-matter experts). We also clarify that our work contributes an evaluation benchmark to assess existing language agents rigorously, which has minimal or no risk in "inadvertently synthesizing toxic or dangerous chemicals."

We have provided detailed responses to other concerns and questions of this reviewer, including the relationship between data-driven discovery and scientific discovery, annotation and evaluation details, and citation formats. We would love to hear back from the reviewer and discuss any remaining concerns.

Sincerely,

Authors of Submission12844

---

### Meta-Review · Area_Chair_2xHc · 2024-12-16

**Metareview:**

We recommend the paper to be accepted for Poster.

The contribution seems timely and addresses some concerns in the literature.

Below more details about this contribution.

The paper introduces ScienceAgentBench, a new benchmark for evaluating language agents for data-driven scientific discovery. To ensure the scientific authenticity and real-world relevance of our benchmark, we extract 102 tasks from 44 peer-reviewed publications in four disciplines and engage nine subject matter experts to validate them.

The key strengths (S#) of the paper are as follow:

- (S1)	The paper creatively applies language agents to new domains, filling a gap where existing benchmarks fall short.
- (S2)	The benchmark is rigorously developed with input from nine subject matter experts, ensuring tasks are authentic and challenging.
- (S3)	The authors proactively mitigate data contamination by modifying datasets, enhancing the reliability of their evaluation. They use comprehensive evaluation metrics—including Valid Execution Rate (VER), Success Rate (SR), CodeBERTScore (CBS), and computational costs—to provide a holistic assessment of agent performance.
- (S4)	The study involved extensive data curation and human annotation, demonstrating the authors' dedication and thoroughness. The inclusion of both end-to-end and fine-grained metrics allows for a comprehensive evaluation of models, particularly when the models can only partially solve a problem. Additionally, the exploration and discussion of various interaction methods with the local environment provides valuable insights.

The key weaknesses (W#) of the paper are as follows:

- (W1)	The paper evaluates agents using three frameworks but doesn't justify these choices or explore advanced architectures like ReAct or Toolformer. Without including state-of-the-art frameworks that offer advanced reasoning and tool-use capabilities, the study may not fully assess the agents' potential to handle complex scientific tasks. Incorporating such frameworks could provide deeper insights into their capabilities and limitations.
- (W2)	Human evaluators who also participated in data collection may introduce bias due to familiarity with the tasks, affecting the objectivity of the assessments. Additionally, the error analysis lacks depth, as specific failure modes are not thoroughly examined. Involving independent evaluators and conducting a detailed error analysis would improve objectivity and help identify areas where agents struggle.
- (W3)	The paper doesn't compare the agents' performance with traditional methods or domain-specific tools, making it difficult to assess their practical utility relative to existing solutions. Including such comparisons would provide valuable context to evaluate the agents' real-world usefulness and guide future improvements.
- (W4)	Providing expert domain knowledge doesn't consistently improve agent performance and sometimes even decreases it, suggesting agents struggle to integrate this information effectively. Exploring why agents fail to benefit from expert knowledge could lead to better integration strategies and enhance their overall performance.
- (W5)	Coding generation-related tasks may not be representative of some other scientific domains. While recent research has focused on such tasks, the authors could briefly acknowledge this limitations, especially since the benchmark's name suggests a more comprehensive evaluation of broader scientific capabilities.

We note that the authors addressed many of the reviewers concerns.

**Additional Comments On Reviewer Discussion:**

The authors have been proactive in addressing the comments raised by the reviewers, and the reviewers were well engaged responding to the authors.

No ethics review raised by the reviewers, and we agree with them.

---

### Decision · Program_Chairs · 2025-01-22

Accept (Poster)